# Neurodevelopmental Outcomes Associated with Early-Life Exposure to Heavy Metals: A Systematic Review

**DOI:** 10.3390/ijerph22081308

**Published:** 2025-08-21

**Authors:** André Soares da Silva, Renata Maria Silva Santos, Patricia Gazire De Marco, Victhor Hugo Martins Rezende, Tamires Coelho Martins, Joyce Romano Silva, Marco Aurélio Romano-Silva, Débora Marques de Miranda

**Affiliations:** 1Graduate Program in Molecular Medicine, Faculty of Medicine, Federal University of Minas Gerais (UFMG), Belo Horizonte 30130-100, Brazil; andreso@ufmg.br (A.S.d.S.); patriciademarco@ufmg.br (P.G.D.M.); vhmrezende@sga.pucminas.br (V.H.M.R.); tamires.martins.1364879@sga.pucminas.br (T.C.M.); romanojoyce@ufmg.br (J.R.S.); 2Department of Psychiatry, Faculty of Medicine, Federal University of Minas Gerais (UFMG), Belo Horizonte 30130-100, Brazil; romano-silva@ufmg.br; 3Department of Pediatrics, Faculty of Medicine, Federal University of Minas Gerais (UFMG), Belo Horizonte 30130-100, Brazil; deborammiranda@medicinaufmg.br

**Keywords:** heavy metals, neurodevelopmental, children, lead, mercury, cadmium, arsenic

## Abstract

Introduction: Globally, approximately 53 million children under the age of five live with some form of developmental disability. Exposure to heavy metals has been identified in the literature as a contributing factor in the etiology of neurodevelopmental disorders, however it usually is understudied. Even at low concentrations, these toxicants pose a risk to neurodevelopment, when affecting children early as in the prenatal period. This study aims to systematically review the literature on the associations between exposure to toxic heavy metals and neurodevelopmental outcomes in children. Method: The review was registered at the International Prospective Register of Systematic Review-PROSPERO, under number CRD420250653229 and searches were conducted in the PubMed, Scopus, Web of Science, EMBASE, Lilacs and PsycInfo databases. Results: A total of 68 articles were included, comprising 48 longitudinal studies and 20 cross-sectional studies, published between 2006 and 2025, with a combined sample of 215,195 individuals from 23 countries. Lead was the most consistently investigated metal, appearing in 75% of the studies, followed by mercury, cadmium, and arsenic. Most findings referred to prenatal exposure. Cognitive and motor outcomes were predominantly affected by exposure to Pb and Hg, while behavioral outcomes showed negative associations mainly with Pb and As. Conclusions: The majority of the studies analyzed indicated adverse effects resulting from exposure to heavy metals during pregnancy, especially in the early months, highlighting the vulnerability of the developing brain.

## 1. Introduction

Contamination by heavy metals such as arsenic (As), lead (Pb), cadmium (Cd), and mercury (Hg) poses a serious threat to human health, according to the World Health Organization [1]. These metals accumulate in the food chain, water, and dust, becoming sources of exposure through ingestion, inhalation, or dermal contact [2]. Although dermal absorption of heavy metals is generally low [3], it is especially relevant for children, whose skin is thinner, more permeable, and less resistant to irritants and infections [4]. Children are particularly vulnerable because they ingest and absorb more Pb per unit of body weight than adults [5], and they often engage in risk behaviors such as putting objects or their hands into their mouths [2]. Since early childhood is a critical period for neurological development, exposure to heavy metals may affect the immature nervous system and contribute to neurodevelopmental disorders [6,7].

Child neurodevelopment is broadly assessed across cognitive, language, motor, and social domains. Delays in any of these areas indicate developmental progress below what is expected for a given age group [8]. Neurobehavioral development, which involves brain maturation processes responsible for regulating behavior, cognition, and emotions, is essential for academic performance and psychological well-being [9]. It is estimated that neurobehavioral disorders affect between 10% and 15% of children worldwide, with a significant global rise in the prevalence of autism spectrum disorder (ASD) and attention-deficit/hyperactivity disorder (ADHD) [10,11]. The proper maturation of the central nervous system requires adequate levels of essential elements such as copper (Cu), zinc (Zn), iron (Fe), and selenium (Se) [12]. In contrast, transition metals like manganese (Mn) and heavy metals are toxic even at low concentrations and can impair neurodevelopment beginning in the prenatal period [13].

The literature highlights exposure to heavy metals as a contributing factor in the etiology of neurodevelopmental disorders [14]. However, the magnitude of this association has not yet been fully established, as determining causality and linking specific agents remains challenging [15]. According to comprehensive estimates from the United Nations Children’s Fund (UNICEF), 10.1% of individuals aged 0 to 17 years worldwide have moderate to severe disabilities, with children aged 0 to 4 years accounting for 12.2% of this total [16] (Olusanya et al., 2022). Additionally, it is estimated that 200 million children under the age of five in developing countries are not reaching their full neurodevelopmental potential [17].

Pb is one of the most widely studied neurotoxins and is associated with intellectual and motor deficits, behavioral changes, and mood regulation dysfunctions due to its interference with gene expression during critical periods of brain plasticity [18,19]. Its occupational sources are not yet clearly defined, and most studies focus on specific regions [20]. Areas near mining operations pose elevated risks, with blood Pb levels higher than those in other mining zones, and residues from these activities may continue to contaminate the environment and food chain even after mining has ceased [21,22,23]. Other sources include both formal and informal activities such as battery manufacturing, radiator repair, artisanal production, and the use of lead in fishing weights, jewelry, decorative items, and glazes, potentially affecting women and children [20]. Additionally, personal hygiene products and cosmetics may contain Pb, As, and Hg [24,25].

Exposure to heavy metals during pregnancy has been associated with various neurodevelopmental outcomes. Pb has been linked to an increased incidence of ASD [26], while Cd appears to negatively affect overall neurodevelopment and has also been associated with epilepsy [13]. Additionally, exposure to Hg and As has been connected to structural alterations in the frontal and subcortical brain regions, as well as to dysregulation of neurotransmission [7].

Given the multiple environmental sources of heavy metals and the associated risks of exposure during childhood, several systematic reviews have investigated specific relationships between individual metals and outcomes such as ADHD [27] (Gu et al., 2024), ASD [28,29], cognitive performance [30], and general health problems [31,32]. While these reviews reflect growing concern about the impact of heavy metals on child development, an updated and comprehensive synthesis is needed to organize scattered evidence, integrate different types of exposure, and clarify associations across various neurodevelopmental domains, including motor, cognitive, sensory, social, and behavioral. This review aims to fill this gap by systematically gathering evidence on the association between heavy metal exposure and impairments or, in some cases, advantages in neurodevelopmental performance domains.

## 2. Method

This review was conducted in compliance with Preferred Reporting Items for Systematic Reviews and Meta-Analyses–PRISMA, a protocol that standardizes the collection of evidence [33] and was registered at the International Prospective Register of Systematic Review–PROSPERO, under number CRD420250653229. A search was performed on 15 February 2025 with the following question: What are the associations between exposure to heavy metals and neurodevelopment in children? The searches were conducted in the PubMed, Scopus, Web of Science, EMBASE, Lilacs and PsycInfo databases, using the following descriptors: children AND “heavy metals” AND neurodevelopment. The search strategy is described in detail in Appendix A. Following the PECO strategy, the study population consisted of children, and exposure was represented by the presence of heavy metals in biological samples from mothers during pregnancy or lactation or in children after birth, including dosages in the placenta and umbilical blood cord. Control groups may be absent in observational studies, or represented by unexposed populations, or with different levels of exposure. Outcomes will be assessed by general developmental milestones, or neurodevelopmental disorders. For the purposes of this systematic review, neurodevelopment refers to the structural and functional processes involved in the formation of the central nervous system from gestation, which result in the development of sensory, motor, cognitive, language, social, and behavioral functions in children after birth [34,35].

### 2.1. Inclusion Criteria

Studies of children with or without a diagnosis of neurodevelopmental disorders will be included. Studies in English, with no restriction on publication date.

### 2.2. Exclusion Criteria

Studies on neurodegenerative diseases in childhood were excluded. Studies on neurodevelopmental disorders were excluded if they did not report diagnoses based on validated instruments. According to the Diagnostic and Statistical Manual of Mental Disorders, Fifth Edition, Text Revision (DSM-5-TR), neurodevelopmental disorders begin in the early stages of life and generally follow a stable course, as seen in intellectual disability or ASD [36]. In contrast, pediatric neurodegenerative diseases have a progressive and irreversible course, characterized by the loss of previously acquired neurological skills, as observed in leukodystrophies and lysosomal storage diseases [37]. By “validated instrument” for the diagnosis or assessment of neurodevelopment, we refer to standardized tools with adequate psychometric properties (such as validity and reliability) that are widely accepted in the scientific literature, for example, the Bayley Scales of Infant and Toddler Development (Bayley-III or Bayley-4), used to assess cognitive, motor, and language development in children aged 1 to 42 months. This criterion aims to ensure the inclusion of only studies employing rigorous assessments through standardized instruments, excluding those based solely on expert opinion or informal reports from parents and caregivers.

Studies in which exposure to heavy metals was not demonstrated by validated individual biomonitoring measures were also excluded. Investigations of co-exposures that did not allow extraction of specific data on heavy metals were excluded. This criterion aims to minimize potential confounding factors by excluding studies that assess other toxic elements besides heavy metals, as well as nutrients that can be replenished through supplementation, such as iron. Studies focused on trace element supplementation or iron deficiency anemia, as well as markers of iron metabolism such as transferrin and ferritin, were excluded. Reviews of all types, case reports, case series, gray literature, letters, editorials, and scientific meeting documents were also excluded.

### 2.3. Screening Procedure

The screening procedure was performed in pairs. After exclusion of duplicate records, the titles and abstracts of each study were screened according to the inclusion and exclusion criteria by two reviewers independently. Articles eligible for full reading were separated by each reviewer. Articles that were selected by only one of the two reviewers were considered conflicts and were included or excluded by consensus with a third reviewer. Articles that were not included after full reading had the reason for exclusion stated in the PRISMA flowchart.

### 2.4. Data Extraction

Data were extracted using a standard form that included: first author, date, country of publication, study design, sample characterization (mother and/or child), sample and method used to measure heavy metals, neurodevelopment assessment instruments and main results found.

### 2.5. Quality Assessment

The quality of the studies was assessed using the Joanna Briggs Institute (JBI) Critical Appraisal Tool. The evaluation was performed in pairs, and discrepancies were resolved by consensus with a third reviewer. The higher the sum of the points on the scale, the higher the methodological quality, which can go up to 8 for cross-sectional studies or 11 points for cohort studies. The specific appraisal questions used are available in Appendix A. The study should clearly describe the exposure measurement method and clarify any potential confounding factors that may influence the interpretation of the results. Additionally, measurements should be conducted using validated instruments, and the statistical method used should be the most appropriate [38]. The available guidelines for applying the JBI tool do not specify a cutoff point for determining whether a study is of “Good”, “Fair”, or “Low” quality. Given the nature of the review and the inclusion of studies, mostly with a cross-sectional design, the team placed greater emphasis on criteria assessing risks of selection bias, measurement bias, and confounding bias, specifically questions 1, 3, 4, 5, 6, and 7. For cohort studies, importance was also given to follow-up data and questions 1, 3, 4, 5, 6, 7, 8 and 9 had greater weight. Studies classified as “Good” quality must have 100% positive responses (“Yes”) to these key items.

## 3. Results

### 3.1. Characteristics of the Included Studies

The initial search presented a total of 686 studies. In total, 123 full-text articles were reviewed, and 68 articles met the inclusion criteria. The study selection process is detailed in the PRISMA flowchart, in Figure 1. The included studies were distributed into 48 longitudinal and 20 cross-sectional designs, published between 2006 and 2025. The study samples represented data from 23 countries, comprising a total of 215,195 individuals. From this perspective, Japan accounted for the largest portion of the sample, approximately 181,145 participants, primarily due to two studies [39,40], which included samples of 80,759 and 96,165 individuals, respectively. Table 1 shows the sample distribution by country.

Regarding gender prevalence in the studies, 47 (69.12%) demonstrated a predominance of the male gender, 17 (25%) reported a predominance of the female gender, and 4 (5.88%) studies did not provide information about gender in their samples.

Due to the variability in how the studies reported the ages of their samples, it was not possible to calculate a representative mean age for the entire population included in this review. Nevertheless, it was observed that most samples were concentrated in early childhood (0 to 2 years), with 24 studies. Subsequently, 17 studies included preschool-aged children (2 to 6 years), 10 studies addressed middle childhood (6 to 9 years), and 7 studies focused on pre-adolescents (9 to 12 years). Additionally, five studies included age ranges spanning both early childhood and preschool, and one study involved participants between preschool and middle childhood. Finally, four studies did not specify the age range of the samples analyzed.

### 3.2. Metal Analysis

The high heterogeneity among the included studies prevented them from being grouped based on the metals analyzed. However, when examining the frequency with which each metal was investigated, Pb emerged as the most frequently studied, appearing in 51 studies (75%). It was followed by Hg (33 studies; 48.53%), Cd (20; 29.41%), As (16; 23.53%), Mn (9; 13.24%). Other metals were evaluated such as cesium-Cs (5; 7.35%), cobalt-Co, barium-Ba, and thallium-Tl (4 studies each; 5.88%); nickel-Ni, uranium-U, strontium-Sr, and rubidium-Rb (3 studies each; 4.41%); tin-Sn, titanium-Ti, and vanadium-V (2 studies each; 2.94%); and finally, aluminum-Al, silver-Ag, tungsten-W, cerium-Ce, lanthanum-La, rhenium-Re, beryllium-Be, platinum-Pt, and bismuth-Bi, each cited in only one study (1.47%).

There was also considerable variation in the type of biological samples analyzed, with some studies using samples from the mother, the child, or both. Regarding maternal samples, 32 studies analyzed blood (47.06%), 10 analyzed urine (14.71%), 7 analyzed hair (10.29%), and 4 analyzed breast milk (5.88%). For child-derived samples, blood was analyzed in 37 studies (54.41%), hair in 10 (14.71%), urine in 7 (10.29%), and nails in only 2 (2.94%). Among all sample types, blood was the most frequently analyzed material, being exclusively examined in 43 studies (63.23%). Urine was used exclusively in 9 studies (13.23%), while hair or a combination of hair and nails was analyzed in 8 studies (11.76%).

Regarding the methodology for metal analysis, 34 studies (50%) employed variations in inductively coupled plasma spectrometry, followed by atomic absorption techniques used in 24 studies (34.29%). Eight studies (11.76%) applied more than one analytical method. Additionally, only one study (1.47%) used the Gold Amalgam Mercury Analyzer, and another (1.47%) used the portable LeadCare II system.

### 3.3. Quality Assessment

The summaries of the quality assessment for cross-sectional and longitudinal studies are presented in Appendix A, according to the items of the ‘Analyse of Cross-sectional and cohort studies’ from the JBI. Of the 20 cross-sectional studies assessed, 13 (65%) were of good quality, 7 (35%) were of fair quality and none was of low quality. Regarding the 48 longitudinal studies assessed, 30 (62.50%) were of good quality, 17 (35.40%) were of fair quality, and 1 (2.1%) was of low quality.

### 3.4. Main Associations

The available data could not be meta-analyzed due to the heterogeneity of exposure assessment methodologies, exposure thresholds, and outcomes assessed, which prevented direct comparison. Therefore, we performed a data synthesis for the consistently investigated heavy metals: Pb, Hg, As, Cd, and Mn. Information on each of the 68 included studies is available in Table 2, and more details are available in Appendix A. The associations between metal exposure and neurodevelopmental outcomes are graphically represented in Figure 2, Figure 3 and Figure 4. Associations indicating impaired neurodevelopment consistently outnumbered those that found a positive effect or even those in which no association was found. In both cross-sectional and longitudinal studies, studies of above-average quality were the majority. Most studies investigated the influence of heavy metal exposure on the neurodevelopment of children during the intrauterine period. Therefore, the evidence generated in this review is more robust regarding prenatal exposure to heavy metals.

Of the 68 included studies, 29 (42.6%) presented sufficient data to estimate the effect size of the associations between heavy metal exposure and neurodevelopmental outcomes. The majority (79.3%) indicated a small effect, while moderate and large effects were observed in 13.8% and 17.2% of the studies, respectively. Some studies assessed more than one outcome. Among these 29 studies, 55.2% were classified as good quality, 41.4% as fair quality, and 3.4% as low quality. The results are presented according to the magnitude of the effect of the associations found with the outcomes, organized by heavy metal and including the quality of each study. Effect size thresholds are determined by the statistical methods used to correlate metal levels with each neurodevelopmental outcome, as reported in each study. These statistics are detailed in Table 2. Given the variety of outcomes assessed and the different statistical metrics applied across studies, the same metal may show associations of varying magnitudes, small, medium, or large, reflecting heterogeneity in the strengths of associations observed both within the same study and across different studies.

#### 3.4.1. Associations with Small Effect Sizes

##### Hg and MeHg

Several neurodevelopmental outcomes have been associated with Hg and MeHg. Two fair-quality studies reported associations between Hg exposure and improved global and motor neurodevelopmental scores [41,42]. Conversely, six studies identified associations between Hg or MeHg exposure and impairments in different domains, including social skills [43], behavioral problems and cognition [44,45], mental development [42], and risk of global neurodevelopmental delay [46,47]. The studies by Myers et al. [43], Marques et al. [41,42], Freire et al. [45], and Nyanza et al. [46] were assessed as of fair quality, while those by Al-Saleh et al. [44,47] were classified as of good quality.

##### Pb

Negative impacts on several neurodevelopmental domains have been associated with Pb in twelve studies. Pb exposure was related to lower cognitive performance [48,49,50,51,52,53], gross motor function [54], higher risk of ADHD [55], sensory processing dysfunction [56], lower mental development [57], lower psychomotor development [58], and lower global neurodevelopment [59]. The studies by Guo et al. [50], Liu et al. [52], and Park et al. [55] were assessed as of fair quality, while the others were classified as of good quality.

##### As, Cd, and Mn

Good to fair quality studies have shown that As exposure was associated with worsening cognition, social skills, motor development, increased risk of ASD, and global neurodevelopmental performance [46,60,61,62]. Small effect size associations for Cd indicated increased risk of ADHD and ASD, as well as worsening motor function, observed in good to fair quality studies [40,60,61]. Exposure to Mn was associated with increased risk of ASD and poorer cognitive performance in good quality studies [61,62].

#### 3.4.2. Associations with Medium Effect Sizes

Unfavorable associations were observed with neurodevelopmental outcomes, particularly with Hg and Pb. Among good-quality studies, Kim et al. [63] found that Hg exposure was associated with poorer mental development. Among fair-quality studies, Al-Saleh et al. [64] reported that higher Hg levels were associated with lower cognitive function scores and a higher risk of visuomotor integration problems. A fair-quality study found worse language performance among children exposed to Hg [46]. Finally, a low-quality study found an association between Pb exposure and lower cognition scores [65].

#### 3.4.3. Associations with Large Effect Sizes

Adverse effects with large effect sizes were identified for Pb, Cd, and Hg, particularly affecting cognition, behavior, and global development. Among good-quality studies, Nie et al. [66] reported that Pb exposure was associated with worse cognitive and behavioral performance. Also in this group, Kim et al. [51] observed that Hg was associated with worse psychomotor performance, and Kao et al. [67] found that Cd exposure was associated with worse cognitive outcomes. Among fair-quality studies, Halabicky et al. [68] found an association between Pb and lower cognitive performance, while Nyanza et al. [46] reported that Cd was related to impairments in global neurodevelopment.

**Table 2 ijerph-22-01308-t002:** Main associations between heavy metals and neurodevelopment.

Author/Year/Country/Type Of Study	N|Age|Gender %|Neurodevelopmental Assessment	Bio Sample Type|Major Associations	Quality
Abd Wahil et al., 2022 Malaysia, Cross-sectional [19]	Typically Develop: 74 Age: 5.45 (0.83) years Males: 39 (52.70%) males ASD: 81 Age: 5.63 (0.60) years Males: 68 (84%) males Neurodevelopmental: ASD diagnoses by DSM-5 criteria and International Classification of Diseases-10 (ICD-10)	Bio sample: Child urine. Urinary Pb levels were significantly lower in children with ASD (mean 0.26 ± 0.31 μg/dL) compared to typically developing children (mean 0.58 ± 0.41 μg/dL) (*p* < 0.05).	***
Al-Saleh(a) et al., 2016 Saudi Arabia, Cross-sectional [44]	N: 944 Age: 7.96 (2.68) months Gender: 487 (51.6%) males Neurodevelopmental: Denver Developmental Screening Test II (DDST-II) Parents’ Evaluation of Developmental Status (PEDS)	Bio sample: Maternal blood, milk and urine/Child hair, blood and urine. Neonatal hair and urinary Hg levels were positively associated with an increased risk of future learning problems (OR = 1.209, 95% CI = 1.001–1.460). Neonatal hair MeHg levels were positively associated with potential delays in overall development (OR = 1.193, 95% CI = 1.044–1.365). Neonatal hair and urinary Hg levels were positively associated with an increased risk of future behavioral problems by Parents Evaluation of Developmental Status scale (OR = 1.337, 95% CI = 1.012–1.767).	***
Al-Saleh(b) et al., 2016 Saudi Arabia, Cross-sectional [47]	N: 944 Idade: 8.0 (2.7) months Gender: 487 (51.6%) males Neurodevelopmental: Denver Developmental Screening Test and II (DDST and DDST-II) Parents’ Evaluation of Developmental Status (PEDS)	Bio sample: Maternal and child Urine and hair/Maternal blood and milk. Maternal hair MeHg levels were positively associated with delayed development risk test outcomes (adjusted OR = 1.193, *p* = 0.01).	***
Al-Saleh et al., 2020 Saudi Arabia, Longitudinal [64]	N: 82 Age: 6.81 (0.64) years Gender: 36 (43.9%) males Neurodevelopmental: Beery-Visual-Motor Integration (Beery VMI) Test Of Non-Verbal Intelligence (TONI)	Bio sample: Maternal blood, hair and urine/Child hair and urine. Urinary Hg levels in infants during breastfeeding were associated with lower visual-motor integration (VMI) scores (ß = −0.469, 95% CI = −7.541, −2.13). Urinary Hg levels in infants during breastfeeding were associated with lower non-verbal intelligence scores (ß = −0.359, 95% CI = −5.857, −0.845). No correlation was found between neurodevelopmental scores and levels of MeHg in hair, Hg in urine, or Pb in urine among children during follow-up.	**
Boucher et al., 2010 Canada, Longitudinal [69]	N: 574 Age: 3–6 years Gender: 305 (53.13%) males Neurodevelopmental: Event-related potentials (ERPs) Electro-oculogram (EOG)	Bio sample: Umbilical cord and venous blood. Umbilical cord blood MeHg concentrations were not associated with children’s neurodevelopmental parameters, in the follow-up.	***
Cai et al., 2019 China, Cross-sectional [56]	N: 48 Age: At birth-6 months Gender: 31 (62%) males Neurodevelopmental: Sensory Processing Measure-Hong Kong Chinese version (SPM-HKC)	Bio sample: Child blood. Children’s blood Pb levels were positively associated with the total score of sensory processing dysfunction (r = 0.168, *p* < 0.01).	***
de Assis Araújo et al., 2021 Brazil Longitudinal [70]	N: 118 Age: 11.3 (0.57) years Gender: 41 (34.7%) males Neurodevelopmental: Denver Developmental Screening Test II (DDST-II).	Bio sample: Maternal and umbilical cord blood. Maternal blood As levels were higher in the group of children with more failures on the global neurodevelopmental test (*p* = 0.03).	**
Deroma et al., 2013 Italy, Longitudinal [71]	N: 154 Age: 7.7 (0.7) years Gender: 77 (50%) males Neurodevelopmental: Wechsler Intelligence Scale for Children (WISC-III)	Bio sample: Maternal hair and milk/Child hair. Total Hg or MeHg levels in hair were not associated with children’s IQ scores.	***
Desrochers-Couture et al., 2018 Canada, Longitudinal [49]	N: 609 Age: 3.4 (0.3) years Gender: 297 (48.8%) males Neurodevelopmental: Wechsler Preschool and Primary Scale of Intelligence (WPPSI-III)	Bio sample: Maternal and umbilical cord blood. Cord blood Pb levels were negatively correlated with IQ in boys (b = −3.28). Umbilical cord blood Pb levels showed no significant correlation with Performance IQ in girls (β = −3.28).	***
Farías et al., 2022 Mexico, Longitudinal [72]	N: 253 Age: 1–12 months Gender: 132 (52.1%) males Neurodevelopmental: Bayley Scales of Infant Development (BSID-III)	Bio sample: Maternal blood Maternal blood Pb levels were negatively associated with offspring language development (*p* < 0.05).	**
Freire et al., 2010 Spain, Cross-sectional [45]	N: 72 Age: 51 months Gender: 72 (100%) males Neurodevelopmental: McCarthy Scales of Children’s Abilities (MSCA) General Cognitive Score (GCS)	Bio sample: Child hair Children’s total hair Hg levels (T-Hg in µg/g) showed negative associations with neurodevelopment at 4 years of age, specifically in the domain of general cognition (β = −2.09, 95% CI = −5.72 to 1.54). Children’s total hair Hg levels (T-Hg in µg/g) showed negative associations with neurodevelopment at 4 years of age in the domains of gross motor skills (β = −1.09, 95% CI: −5.25 to 3.07) and fine motor skills (β = −1.03, 95% CI: −5.46 to 3.41).	**
Guo et al., 2020. China, Longitudinal [50]	N: 326 Age: 7.4 (0.4) years Gender: 186 (57%) males Neurodevelopmental: Chinese Revised-Wechsler Intelligence Scale for Children (C-WISC-IV) Verbal Intelligence Quotient (VIQ) Performance Intelligence Quotient (PIQ) Full Intelligence Quotient (FIQ),	Bio sample: Maternal urine Prenatal maternal urinary Pb levels were negatively associated with children’s total IQ at 7 years of age (β = −2.31, 95% CI: −4.13 to −0.48; *p* = 0.013).	**
Halabicky et al., 2023 China, Longitudinal [68]	N: 417 Age: 11.51 (0.39) years Gender: 220 (52.76%) males Neurodevelopmental: Wechsler Intelligence Scale for Children-Revised (WISC-R) Working Memory Measurement Software (WM)	Bio sample: Child blood Children’s blood Pb levels were associated with lower IQ scores (β = −0.59) at 3 to 5 years of age. No association was found between Pb blood levels and worsened IQ outcomes (β = −0.59) in 12-year-old children.	**
Hu et al., 2006 Mexico, Longitudinal [57]	N: 146 Age: Not applicate Gender: 76 (52.05%) males Neurodevelopmental: Bayley Scales of Infant Development II–Spanish version (BSID-IIS)	Bio sample: Umbilical cord and venous blood. Increased plasma Pb and whole blood Pb levels during the first trimester of pregnancy were associated with lower mental development scores by BSID-II scale at 24 months of age (β = −3.54, *p* = 0.03 and β = −2.4, *p* = 0.19, respectively).	***
Hu et al., 2016 China, Cross-sectional [73]	N: 410 Age: Not applicable Gender: 214 (52.19%) males Neurodevelopmental: Gesell Developmental Schedules (GDS)	Bio sample: Maternal and umbilical cord blood. Umbilical cord blood Hg levels were positively associated with developmental quotients in the adaptive domain by GDS scale (β = 4.22) and the social domain (β = 4.06).	***
Huang et al., 2012 Taiwan, Longitudinal [74]	N: 119 Age: 2–9 years Gender: 62 (52.1%) males Neurodevelopmental: Bayley Scales of Infant Development-II (BSID-II) Wechsler Preschool and Primary Scale of Intelligence-Revised (WPPSI-R) Wechsler Intelligence Scale for Children-version III (WISC-III)	Bio sample: Umbilical cord and venous blood. Children’s blood Pb levels were negatively associated with total IQ scores between ages 5 and 6 years (β = −5.97, SE = 2.59, *p* = 0.025) and between ages 2 and 9 years (β = −0.289, 95% CI: −16.9 to −1.48, *p* = 0.020).	***
Inoue et al., 2022 Japan, Longitudinal [39]	N: 80,759 Age: Not applicable Gender: 41,190 (51%) males Neurodevelopmental: Ages and Stages Questionnaires (ASQ-III)	Bio sample: Maternal and umbilical cord blood. No significant associations were found between Pb levels in umbilical cord blood and suspected neurodevelopmental delay during the first three years of life.	**
Kao et al., 2021 Taiwan, Cross-sectional [54]	N: 139 Age: 2.8 (0.4) years Gender: 66 (47%) males Neurodevelopmental: Bayley Scales of Infant Development-III (BSID-III)	Bio sample: Child hair and nail Children’s hair Pb concentrations were negatively associated with gross motor scores (β = −0.04, 95% CI: −0.07 to −0.01) after adjustment for residence near a highway.	***
Kao et al., 2023 Taiwan, Longitudinal [67]	N: 152 Age: At 24 months Gender: 72 (47.36%) males Neurodevelopmental: Bayley Scales of Infant Development-III (BSID-III)	Bio sample: Child hair and nail. In low birth weight premature children, Cd concentrations in the nails were negatively associated with cognition (β = −0.63, 95% CI: −1.17 to −0.08).	***
Kashala-Abotnes et al., 2016 Democratic Republic of Congo, Cross-sectional [75]	N: 89 Age: 17.5 (4.3) months Gender: 52 (58.4%) males Neurodevelopmental: Baby characteristics questionnaire (BCQ)	Bio sample: Child blood. No statistically significant differences were observed in child neurodevelopment.	***
Kim et al., 2009 South Korea Cross-sectional [51]	N: 261 Age: 9.7 (0.6) years Gender: 141 (54%) males Neurodevelopmental: Korean Educational Development Institute-Wechsler Intelligence Scales (KEDI-WISC)	Bio sample: Venous blood. Pb levels were negatively associated with total IQ (β = −0.174) and verbal IQ (β = −0.187).	**
Kim et al., 2018 South Korea, Longitudinal [63]	N: 1098 Age: 6–36 months Gender: 571 (52%) males Neurodevelopmental: Bayley Scales of Infant Development-II (BSID-II) to evaluate the Psychomotor Development Index (PDI) and Mental Development Index (MDI)	Bio sample: Maternal and umbilical cord blood. Prenatal exposure to Hg in early pregnancy was inversely associated with psychomotor development index (PDI) scores at 6 months (β = −0.550, *p* = 0.031). Prenatal exposure to Hg in early pregnancy was inversely associated with mental development index (MDI) scores at 6 months (β = −0.408, *p* = 0.048).	***
Kippler et al., 2016 Greece, Longitudinal [76]	N: 575 Age: 4.2 (0.23) years Gender: 288 (50%) males Neurodevelopmental: McCarthy Scales of Children’s Abilities (MSCA)	Bio sample: Maternal urine. Maternal urinary Cd concentrations ≥0.8 µg/L during pregnancy were inversely associated with children’s overall cognitive scores at 4 years of age (β = −6.2, 95% CI: −12 to −0.54, *p* = 0.032). Maternal urinary Pb concentrations ≥0.8 µg/L during pregnancy were not associated with children’s overall cognitive scores (β = −6.2; 95% CI: −12, 0.54; *p* = 0.45).	***
Kou et al., 2025 Spain, Longitudinal [77]	N: 400 Age: 40 days Gender: 210 (52.5%) males Neurodevelopmental: Bayley Scales of Infant and Toddler Development-III (BSID-III)	Bio sample: Maternal urine. The mixture of heavy metals: Cd, Ni, Hg, and Pb was associated with poorer expressive language scores (β = −0.26, 95% CI: −0.44 to −0.07). Cd was the most influential contributor to lower cognitive scores (β = −1.47, *p* = 0.044).	***
Lee et al., 2017 South Korea, Longitudinal [78]	N: 251 Age: 6–60 months Gender: Not informed Neurodevelopmental: The Korean version of Bayley Scales of Infant and Toddler Development-II (K-BSID-II) to evaluate the Psychomotor Development Index (PDI) and Mental Development Index (MDI) The Korean version of the Wechsler Preschool and Primary Scale of Intelligence–Revised (K-WPPSI-R)	Bio sample: Umbilical cord blood. Cognitive development stability appeared to be unaffected by heavy metal levels in umbilical cord blood.	**
Lee et al., 2018 Taiwan, Cross-sectional [79]	N:122 Health control: N: 46 Age: 8.1 (1.2) years Gender: 31 (67.4%) males ADHD-I:N: 29 Age: 8.0 (1) years Gender: 11 (37.93%) males TDAH-H/I: N: 47 Age: 7.7 (1) years Gender: 40 (85.10%) males Neurodevelopmental: Schedule for Affective Disorders and Schizophrenia for School-Age Children, epidemiologic version (K-SADS-E) Wechsler Intelligence Scale for Children–Fourth Edition (WISC-IV) Swanson, Nolan, and Pelham Version IV Scale (SNAP-IV)	Bio sample: Child urine. Cd (*p* < 0.05) and Pb (*p* < 0.01) levels were negatively correlated with the IQ. Sb (*p* < 0.01) and Pb (*p* < 0.05) levels were positively correlated with the severity of ADHD symptoms	**
Lee et al., 2021 South Korea, Longitudinal [80]	N: 502 Age: 4–6 years Gender: 254 (50.6%) males Neurodevelopmental: Korean Educational Developmental Institute’s Wechsler Intelligence Scale for Children (KEDI-WISC)	Bio sample: Maternal and child blood. Blood Mn levels at 4 years of age were negatively associated with children’s IQ (β = −5.99, 95% CI: −11.37 to −0.61).	***
Lin et al., 2013 Taiwan, Cross-sectional [81]	N: 230 Age: At birth to 2 years Gender: 128 (55.7%) males Neurodevelopmental: Comprehensive developmental inventory for infants and toddlers (CDIIT)	Bio sample: Umbilical cord blood. General developmental quotients were significantly lower in the group with high exposure to Mn and Pb (β = −7.03, SE = 2.65, *p* = 0.009). High exposure to Mn and Pb was associated with lower cognitive quotients (β = −8.19, SE = 3.17, *p* = 0.011) and language quotients (β = −6.81, SE = 2.73, *p* = 0.013). No significant differences were found in language development quotients (β = −6.81, SE = 2.73, *p* = 0.013) in the group with high exposure to As or Hg. No significant differences were found in cognitive development quotients (β = −8.19, SE = 3.17, *p* = 0.011) in the group with high exposure to As or Hg.	***
Liu et al., 2024 Mexico, Longitudinal [52]	N: 533 Age: 9 (9–10) years Gender: 274 (51.4%) males Neurodevelopmental: Go/NoGo Happy, Go/NoGo Neutral, Go/NoGo Letter and D-KEFS Color Word Interference Test	Bio sample: Umbilical cord blood. Pb concentrations in umbilical cord blood (β = −0.06, 95% CI: −0.11 to −0.01) and in blood at 4 years of age (β = −0.07, 95% CI: −0.12 to −0.02) were negatively associated with inhibitory control in childhood. There was a significant negative association between umbilical cord blood Pb levels and childhood hyperactivity (β = −0.117, *p* = 0.009).	**
Liu J. et al., 2014 China, Longitudinal [58]	N: 243 Age: 6–36 months Gender: 129 (53.09%) males Neurodevelopmental: Bayley Scales of Infant Development-II (BSID-II) to evaluate the Psychomotor Development Index (PDI) and Mental Development Index (MDI)	Bio sample: Umbilical cord blood and venous blood. Significant deficit in psychomotor development index (PDI) due to prenatal Pb exposure at 36 months (β = −1.302, *p* = 0.041). Umbilical cord blood Pb levels were inversely associated with mental development index scores (β = −1.291, *p* = 0.036).	***
Liu J. et al., 2014 China, Longitudinal [59]	N: 332 Age: 3 days Gender: 185 (55.72%) males Neurodevelopmental: Neonatal behavioral neurological assessments (NBNA)	Bio sample: Maternal blood. Infant exposure to Pb during the first trimester of pregnancy was positively associated with reduced neonatal development scores (β = −4.86, 95% CI: −8.831 to −0.889, *p* = 0.03).	***
Llop et al., 2012 Spain, Longitudinal [82]	N: 1.683 Age: 14 (11–23) months Gender: 882 (52.4%) males Neurodevelopmental: Bayley Scales of Infant Development	Bio sample: Umbilical cord blood. No overall significant association was found between Hg levels and neurodevelopment.	***
Llop et al., 2016 Spain, Cross-sectional [83]	N: 1362 Age: 4.8 (0.61) years Gender: 712 (52.3%) males Neurodevelopmental: McCarthy Scales of Children’s Abilities (MSCA)	Bio sample: Umbilical cord blood. Umbilical cord blood Hg concentrations were positively associated with cognitive development scale scores (β = 1.29, 95% CI: 0.28 to 2.31).	***
Lu et al., 2023 China, Longitudinal [84]	N: 275 Age: 18.32 (0.68) months Gender: 140 (50.91) males Neurodevelopmental: Bayley Scales of Infant and Toddler Development-III (BSID-III)	Bio sample: Umbilical cord blood. Umbilical cord blood Pb levels were associated with lower fine motor control scores in girls (β = −1.5, 95% CI: −2.6 to −0.4).	**
Ma et al., 2021 Japan, Longitudinal [85]	N: 3545 Age: At birth to 2 years Gender: 1781 (50.2%) males Neurodevelopmental: Kyoto Scale of Psychological Development (KSPD)	Bio sample: Umbilical cord blood and maternal blood. Elevated Cd concentrations in maternal or umbilical cord blood were not significantly associated with neurodevelopmental delay.	***
Marques et al., 2012 Brazil, Cross-sectional [41]	N: 668 Age: 19.82 (14.47) months Gender: Not informed Neurodevelopmental: Gesell Developmental Scores (GDS)	Bio sample: Child hair. Children’s average Hg concentration was positively correlated with neurodevelopment scores by GDS scale (r = 0.080, *p* = 0.035).	**
Marques et al., 2015 Brazil, Longitudinal [42]	N: 294 Age: 6–24 months Gender: 105 (35.71%) males Neurodevelopmental: Bayley Scales of Infant and Toddler Development-II (BSID-II) to evaluate the Psychomotor Development Index (PDI) and Mental Development Index (MDI)	Bio sample: Child hair. Neonatal hair Hg levels showed a significant negative association with mental development index scores in boys at 24 months (β = −0.222, 95% CI: −0.44 to −0.01, *p* = 0.0451). Neonatal hair Hg levels showed a positive association with age at walking onset in girls (β = 0.188, 95% CI: 0.032 to 0.344, *p* = 0.019).	**
Masumoto et al., 2022 Japan, Longitudinal [40]	N: 96,165 Age: At birth to 3 years Gender: 49,257 (51.2%) males Neurodevelopmental: Ages and Stages questionnaires (ASQ-3)	Bio sample: Maternal blood. Blood Cd concentration was associated with gross motor function delay 1.5 years after birth (adjusted OR = 1.19, 99.7% CI: 1.01–1.40).	***
Merced-Nieves et. al., 2022 Mexico, Longitudinal [86]	N: 549 Age: 6.7 (0.5) years Gender: 278 (50.6%) males Neurodevelopmental: National Center for Toxicological Research (NCTR) Operant Test Battery (OTB)	Bio sample: Maternal blood and umbilical cord blood. Umbilical cord blood Pb levels were associated with altered temporal perception in children (*p* = 0.02).	**
Myers et al., 2004 Seychelles, Longitudinal [43]	N: 643 Age: 8.97 (0.33) years Gender: Not informed Neurodevelopmental: Child Behaviour Checklist (CBCL), WISC	Bio sample: Maternal and child hair. There was a positive association between postnatal Hg exposure and Thought Problems (*p* = 0.011). There was a significant negative association between prenatal Hg exposure and Social Problems scores (β = −0.029).	**
Naspolini et al., 2024 Brazil, Longitudinal [65]	N: 157 Age: 3–16 months Gender: 76 (48.41) males Neurodevelopmental: Bayley Scales of Infant and Toddler Development-III (Bayley-III).	Bio sample: Breast milk. Infants exposed to Pb showed significantly lower language performance at 10–16 months of age (β = −0.413; 95% CI: −0.653 to −0.173) compared to unexposed infants.	*
Nie et al., 2011 USA, Cross-sectional [66]	N: 11 Age: 11 (1.6) years Gender: 55% males Neurodevelopmental: Wechsler Intelligence Scale for Children–Fourth Edition (WISC-IV) Behavior Rating Inventory of Executive Function (BRIEF) Conners ADHD/DSM-IV Scale (CADS-IV)	Bio sample: Blood. Blood Pb levels were associated with increased externalizing problems, internalizing problems, and behavioral symptoms (r = 0.943, r = 0.648, and r = 0.853, respectively). Blood Pb levels showed a positive association with Verbal Comprehension, Working Memory, and IQ (r = 0.746, r = 0.853, and r = 0.823, respectively).	***
Notario-Barandiaran et al., 2024 Spain, Cross-sectional [87]	N: 962 Age: 4.45 (4–6.4) years Gender: 502 (52.2%) males Neurodevelopmental: McCarthy Scales of Children’s Abilities (MSCA)	Bio sample: Urine. Exposure to the mixture of Cu Se, Pb, and Zn was associated with poorer verbal executive function (β = −1.88, 95% CI: −3.17 to −0.59). Exposure to the mixture of inorganic and organic As was associated with poorer gross motor function (β = −1.41, 95% CI: −2.36 to −0.46).	***
Nozadi et al., 2021 USA, Cross-sectional [60]	N: 327 Age: 10–13 months Gender: 163 (49.8%) males Neurodevelopmental: Ages and Stages Questionnaire Inventory (ASQ:I)	Bio sample: Maternal urine and blood. Problem-solving ability showed negative correlations with urinary Mo, Sb, and As levels (r = −0.001, r = −0.106, and r = −0.124, respectively). Fine motor skills showed negative correlations with urinary Ba and As levels (r = −0.132 and r = −0.156, respectively). Social behavior showed negative correlations with urinary As and urinary U levels (r = −0.100 and r = −0.01, respectively). Fine motor skills showed positive correlations with blood Cd, urinary Cs, urinary Mo, urinary Sr, and urinary W levels (r = 0.070, r = 0.038, r = 0.065, r = 0.021, and r = 0.120, respectively). Gross motor skills showed a positive correlation with urinary Sr levels (r = 0.113). Urinary W levels were positively correlated with communication development (r = 0.136). Problem-solving ability showed positive correlations with urinary Cs, Sn, Sr, Tl, and W levels (r = 0.039, r = 0.073, r = 0.089, r = 0.120, and r = 0.079, respectively).	**
Nyanza et al., 2021 Tanzania, Longitudinal [46]	N: 439 Age: 7.92 (1.77) months Gedner: 213 (48.6%) males Neurodevelopmental: Malawi Developmental Assessment Tool (MDAT).	Bio sample: Maternal blood and urine. Higher prenatal As exposure was positively associated with an increased risk of social impairment in girls (adjusted prevalence ratio [aPR] = 1.01, 95% CI: 1.00–1.02, *p* < 0.05). Higher prenatal Hg exposure was positively associated with global neurodevelopmental impairment in girls (adjusted prevalence ratio [aPR] = 1.10, 95% CI: 1.06–1.14, *p* < 0.001). Prenatal exposure to As and Cd was positively associated with global neurodevelopmental impairment in boys (adjusted prevalence ratios [aPR] = 1.03, 95% CI: 1.02–1.048, *p* < 0.001; and aPR = 2.55, 95% CI: 1.33–4.87, *p* < 0.05, respectively). Higher prenatal Hg exposure was positively associated with language impairment in boys (adjusted prevalence ratio [aPR] = 1.05, 95% CI: 1.04–1.07) and girls (aPR = 1.21, 95% CI: 1.16–1.27, *p* < 0.001).	**
Orenstein et al., 2014 USA, Cross-sectional [88]	N: 393 Age: 8.1 (0.6) years Gender: 196 (49.9%) males Neurodevelopmental: Wide Range Assessment of Memory and Learning (WRAML)	Bio sample: Maternal hair. The figure memorization capacity index (visual memory) was negatively associated with MeHg concentration (β = −3.1).	***
Parajuli et al., 2013 Nepal, Cross-sectional [89]	N = 100 Age: 17.4 (3.3) hours Gender: Not informed Neurodevelopmental: Brazelton neonatal behavioral assessment scale (NBAS III)	Bio sample: Umbilical cord blood. Pb and As levels in umbilical cord blood were negatively associated with motor function scores (β = −2.29, 95% CI: −4.35 to −0.24; β = −3.03, 95% CI: −6.05 to −0.01, respectively) at neonatal stage.	**
Parajuli et al., 2014 Nepal, Cross-sectional [90]	N: 100 Age: At birth-6 months Gender: 47 (47%) males Neurodevelopmental: Bayley Scale of Infant Development (BSID II)	Bio sample: Umbilical cord blood. There was no association between Pb levels and infant development scale scores at 6 months.	***
Parajuli et al., 2015 Nepal, Longitudinal [91]	N: 100 Age: 36.9 (0.4) months Gender: 47 (47%) males Neurodevelopmental: Bayley Scale of Infant Development, Second Edition (BSID II)	Bio sample: Umbilical cord blood. Cord blood toxic element levels (Pb and As) were not associated with any developmental delay scores at 36 months.	***
Park et al., 2016 South Korea, Cross-sectional [55]	ADHD sample: 114 Age: 8.79 (1.57) years Gender: 83 (72.8%) males Controls sample: 114 Age: 8.73 (1.65) years Gender: 81 (71.1%) males Neurodevelopmental: Kiddie-Schedule for Affective Disorders and Schizophrenia Present and Lifetime Version (K-SADS-PL-K) Continuous Performance Test (CPT)	Bio sample: Child blood. Children with ADHD showed significantly higher blood Pb concentrations compared to controls (*p* = 0.003). Total blood Pb concentration was associated with an increased risk of ADHD (odds ratio [OR] = 1.60, 95% CI: 1.04–2.45, *p* < 0.05).	**
Polanska et al., 2018 Poland, Longitudinal [92]	N: 402 Age: At birth-24 months Gender: 195 (48.5%) males Neurodevelopmental: Bayley Scales of Infant and Toddler Development-III (BSID III)	Bio sample: Maternal and umbilical cord blood. Higher umbilical cord blood Pb levels were associated with lower cognitive function scores in boys (β = −2.07, 95% CI = −4.07 to −0.06, *p* = 0.04).	**
Qiu et al., 2024 China, Longitudinal [93]	N: 854 Age: 36 (0.3) months Gender: 416 (48.7%) males Neurodevelopmental: Bayley Scales of Infant and Toddler Development- III (BISD-III)	Bio sample: Maternal urine. Higher levels of the metal mixture—V, Cu, Zn, Sb, Ce, and U—were associated with a 2.37-fold increased risk of suboptimal gross motor development (95% CI: 1.15–4.86, *p* = 0.012).	***
Rodrigues et al., 2016 Bangladesh, Longitudinal [48]	N: 525 Age: 2.3 (1.7–3.3) years Gender: 261 (47,9%) males Neurodevelopmental: Bayley Scales of Infant and Toddler Development, Third Edition (BSID-III)	Bio sample: Child blood. Pb levels were negatively associated with cognitive development (β = −0.17).	***
Rosa et al., 2024 USA, Longitudinal [94]	N: 326 Age: 6.53 (2.15) years Gender: 182 (55.8%) males Neurodevelopmental: NIH Toolbox Cognition Battery (NIHTB-CB).	Bio sample: Maternal urine. There was no evidence of an association between the mixture of metals and overall cognitive functioning scores in children.	**
Rothenberg et al., 2016 China, Longitudinal [95]	N: 270 Age: At birth-12 months Gender: 127 (47%) males Neurodevelopmental: Bayley Scales of Infant Development (BSID)-II.	Bio sample: Maternal blood and hair. Higher hair Hg concentration was negatively associated with the mental development index (β = −4.9). Higher blood Pb concentration was negatively associated with the psychomotor development index (β = −11).	***
Ruiz-Castell et al., 2012 Bolivia, Longitudinal [96]	N: 246 Age: 10.5–12.5 months Gender: 129 (52.44%) males Neurodevelopmental: Bayley Scales of Infant Development (BSID)	Bio sample: Maternal blood. Maternal blood Pb levels were positively associated with mental development indices (β = 2.27, *p* = 0.034). Maternal blood cesium levels were positively associated with infant psychomotor development (β = 2.20, *p* = 0.015).	***
Shah-Kulkarni et al., 2020 South Korea, Longitudinal [97]	N: 523 Age: At birth-6 months Gender: 283 (54.1%) males Neurodevelopmental: Korean version of Bayley Scales of Infant Development II (KBSID-II)	Bio sample: Maternal and umbilical cord blood. Exposure to mixtures of Pb, Hg, and Cd in early pregnancy and umbilical cord blood did not significantly affect mental development scores at 6 months. Exposure to mixtures of Pb, Hg, and Cd in early pregnancy and umbilical cord blood did not significantly affect psychomotor development scores at 6 months.	**
Shekhawat et al., 2021 India, Longitudinal [98]	N: 167 Age: At birth-6 months Gender: 80 (48%) males Neurodevelopmental: The Bayley Scale of Infants Developments-III (BSID-III)	Bio sample: Umbilical cord blood. Umbilical cord blood Pb concentrations between 5.0 and 10.5 μg/dL were negatively associated with gross motor skills subscale scores (β = −0.29, 95% CI: −5.00 to 0.11, *p* = 0.042) at a mean age of 6.5 months.	***
Skogheim et al., 2021 Norway, Longitudinal [61]	ADHD: 705 Age: From 2 years Gender: 520 (73.8%) males ASD: 397 Age: From 2 years Gender: 336 (84.6%) males Controls: 1034 Age: From 2 years Gender: 705 (68.2%) males Neurodevelopmental: Adult ADHD Self-Report Scale	Bio sample: Maternal blood. The risk of ADHD in children exposed to the highest quartile of Cd compared to the lowest quartile was increased [OR = 1.59 (95% CI: 1.15, 2.18)]. Elevated risk of ASD was observed in children in the second quartile of As [OR = 1.77 (95% CI: 1.26, 2.49)] and in the highest quartiles of Cd [OR = 1.57 (95% CI: 1.07, 2.31)] and Mn [OR = 1.84 (95% CI: 1.30, 2.59)] compared to the first quartile (reference)	***
Tatsuta et al., 2014 Japan, Longitudinal [99]	N: 387 Age: 42.1 (40–45) months Gender: 202 (52.2%) males Neurodevelopmental: Kaufman Assessment Battery for Children (K-ABC)	Bio sample: Umbilical cord blood. No significant associations were found between prenatal exposure to total Hg (THg) or Pb and children’s intelligence scores.	***
Tatsuta et al., 2020 Japan, Longitudinal [53]	N: 289 Age: 12 (11.1–12.8) years Gender: 148 (51.21%) males Neurodevelopmental: Wechsler Intelligence Scale for Children-Fourth Edition (WISC-IV) Boston Naming Test (BNT)	Bio sample: Umbilical cord and venous blood. Among boys, IQ was associated with child-blood Pb (B = −16.362, *p* = 0.033), but there was no association with Pb in cord blood (B = −6.844, *p* = 0.309). Umbilical cord blood Pb levels showed no significant correlation with any developmental outcomes among girls.	***
Taylor et al., 2018 United Kingdom, Longitudinal [100]	N: 1558 Age: 7 years Gender: 780 (50.06%) males Neurodevelopmental: ALSPAC Coordination Test	Bio sample: Maternal blood. There was no evidence of associations between prenatal exposure to Pb, Cd, or Hg and motor skills measured at 7 years of age. Furthermore, no associations were found with probable developmental coordination disorder.	***
Tong et al., 2022 China, Longitudinal [101]	N: 2164 Age: 55.6 (6.9) months Gender: 1117 (51.6%) males Neurodevelopmental: Wechsler Preschool and Primary Scale of Intelligence-Fourth Edition (WPPSI-IV).	Bio sample: Maternal blood and umbilical cord blood. Higher maternal serum Tl levels during pregnancy were associated with lower IQ in children (β = −1.51, 95% CI: −2.68 to −0.35, *p* = 0.01).	***
Tong et al., 2023 China, Longitudinal [102]	N: 2164 Age: 4.6 (0.6) years Gender: 1117 (51.6%) males Neurodevelopmental: Chinese version of the Wechsler Preschool and Primary Scale of Intelligence-Fourth Edition (WPPSI-IV)	Bio sample: Maternal blood and umbilical cord blood. Elevated maternal serum Ba levels during pregnancy were significantly associated with reduced childhood intellectual function, as indicated by lower global IQ scores [−3.76 (95% CI: −6.19, −1.33)].	***
Valent et al., 2013 Italy, Longitudinal [103]	N: 606 Age: 0–18 months Gender: 307 (50.7) males Neurodevelopmental: Bayley Scales of Infant and Toddler Development-III (BSID-III).	Bio sample: Maternal hair/blood and umbilical cord blood. Maternal total hair Hg levels were positively associated with language development scores in girls (β = 1.5291, *p* = 0.0445). Total Hg in maternal hair was not associated with positive scores on the language development scale in boys (β = 0.3551, *p* = 0.6278).	**
Valeri et al., 2017 India, Longitudinal [62]	N: 825 Age: At birth-40 months Gender: 419 (50.78%) males Neurodevelopmental: Bayley Scales of Infant and Toddler Development-III (BSID-III)	Bio sample: Umbilical cord blood. Higher levels of Mn and As were associated with decreased cognitive scores (β = −0.206, 95% CI: −0.39 to −0.02, *p* < 0.05).	***
Vejrup et al., 2018 Norway, Longitudinal [104]	N: 2239 Age: At birth-5 years Gender: 1187 (53%) males Neurodevelopmental: Ages and Stages Communication scale (ASQ) Neurodevelopmental: Speech and Language Assessment Scale (SLAS) Language-Related Difficulties list (language 20)	Bio sample: Maternal blood. The analysis showed that maternal blood Hg concentration was not significantly associated with language and communication scale scores.	***
Xue et al., 2020 China, Longitudinal [105]	N: 456 Dyslexic: 228 Age: 9.76 (1.29) years Gender: 171 (75%) males Non-dyslexic: 228 Age: 9.69 (1.25) years Gender: 171 (75%) males Neurodevelopmental: Dyslexia Checklist for Chinese Children (DCCC) Pupil Rating Scale-Revised Screening (PRS)	Bio sample: Child urine. Children with dyslexia showed higher concentrations of Sr (*p* = 0.028), Ag (*p* = 0.014), and U (*p* = 0.005).	***

Good quality: ***, Fair quality: ** and Low quality: *.

## 4. Discussion

The findings of this systematic review indicate that exposure to heavy metals is predominantly associated with impairments in child neurodevelopment. Most studies report adverse effects, although there is variability in the magnitude of these effects and the domains affected. Among the metals analyzed, Pb presents the most consistent evidence, followed by Hg, As, Cd, and Mn. Most associations involve deficits in the cognitive domain, followed by behavioral and motor outcomes, suggesting that different neurobiological pathways may be involved. A recurring theme in good-quality studies is the greater vulnerability of the nervous system during pregnancy, especially in the first trimester. This period shows more robust associations, supporting the hypothesis of critical windows in fetal development [106,107]. The plasticity and immaturity of the developing brain make it particularly susceptible to adverse environmental influences, including persistent and cumulative chemical exposures [108].

Previous studies have identified Pb, Hg, As, and Cd as the main metals associated with neurodevelopmental impairments, with Pb’s adverse effects being particularly consistent [109]. Recent evidence further supports these findings, especially regarding cognitive outcomes [15], and studies focusing on lead report greater impairments in childhood neurobehavior [110]. The meta-analysis by Rosenauer et al. [111] found that high blood Pb levels increase the risk of ADHD, linked to neuronal alterations such as synapse reduction and dysfunctions in neurotransmission and calcium signaling. These changes affect key brain structures involved in attention and behavioral control and involve dysfunctions in the dopaminergic and noradrenergic systems. Inhibitory control, a cognitive function essential for regulating behavior, is compromised in ADHD and was also impaired in participants in the study by Desye et al. [110], reinforcing the strong association between Pb exposure and cognitive deficits identified in our review.

The neurotoxic effects of heavy metals, however, do not occur in isolation. Several biological and physiological factors can modulate their toxicity depending on individual characteristics. One such modulator is the gut microbiota, whose composition influences the bioavailability of metals and can alter their absorption and accumulation in the body [112]. Experimental studies have shown that mice on dietary alterations, such as gluten-free diets, accumulate more metals in their tissues [113], highlighting the microbiome’s role. Because this system also participates in neural maturation, its dysfunction can impact neurodevelopment through the gut–brain axis [114]. The interaction between metals and microbiota is bidirectional: while metals affect microbial composition, the microbiota can either mitigate or intensify the toxic effects of these elements [112]. Evidence also suggests that alterations in the microbiome relate to behavioral manifestations, as observed in mice that received microbiota from individuals with ASD [115]. The findings of this review reinforce the association of ASD and ADHD with elevated levels of metals such as Pb, Cd, As, and Mn [19,55,61,79], indicating that shared neurochemical and developmental pathways may be involved in the etiology of these disorders [61].

Although many studies highlight the negative effects of heavy metals on neurodevelopment, ten articles in this review reported some positive effects, particularly related to Hg [41,43,66,73,83,103]. A common explanation for these findings is the higher fish consumption among the studied populations. While heavy metals alone impair neurocognition, their presence, together with seafood intake, may be associated with cognitive benefits [116,117]. The review by Hibbeln et al. [117] suggests that fish consumption may mitigate the harmful effects of heavy metals, and Lyall et al. [118] found that prenatal fish consumption may reduce the likelihood of ASD diagnoses, whereas low consumption was associated with an increased risk [119], although the latter study acknowledges that such effects cannot be attributed solely to nutrients.

Consistent with these findings, a recent review on seafood consumption and child neurodevelopment found that children aged 10 months to 15 years who consumed fish had better cognitive outcomes [120]. Considering that nutrition from the prenatal period through the first two years of life is widely recognized as crucial for neurodevelopment [121], these findings support the hypothesis that the nutritional benefits of fish may, in part, offset the risks associated with exposure to certain heavy metals. Nevertheless, nutritional deficiencies may amplify the adverse effects of heavy metals such as Pb, reducing the potential benefits of fish consumption [121].

Although most studies included in this review identified associations between heavy metal exposure and impaired child neurodevelopment, effect sizes were established in less than half of them. Furthermore, most of these associations showed small effect sizes, which limits the robustness of the conclusions, as small effects may reflect fragile associations susceptible to bias and methodological variations. However, a considerable number of studies reported moderate to large effect sizes, indicating that in some cases the impacts may be more pronounced. This variation suggests that factors such as exposure levels and duration, environmental interactions, and individual characteristics can influence the magnitude of neurotoxic effects. Moreover, although small effect sizes are often considered less clinically relevant, they can have significant public health implications due to the widespread and cumulative exposure to these substances.

Another important aspect concerns the methodological quality of the studies. Most studies included in this review had a longitudinal design, which strengthens the findings by enabling analysis of temporal relationships between exposure and neurodevelopmental outcomes. Among studies with data available to estimate effect sizes, the majority were classified as good quality, including many that found small associations. This supports the idea that smaller effects are not necessarily due to bias or methodological flaws, but may reflect genuine, subtle effects of these exposures. Conversely, studies reporting moderate and large effects ranged from good to fair quality, indicating that a larger effect size alone does not guarantee greater reliability. Therefore, combining effect size assessment with study quality is essential for properly interpreting the available evidence. This underscores the need for future research investigating mechanisms, specific exposures, and potential moderators to clarify the clinical relevance of these findings.

## 5. Conclusions

This systematic review highlights a strong link between exposure to heavy metals, especially Pb, Hg, As, and Cd, and impaired neurodevelopment in children. The majority of studies analyzed reported adverse effects, particularly when exposure occurred during pregnancy, emphasizing the heightened vulnerability of the developing brain. Cognitive function was the most commonly evaluated domain, with Pb consistently showing the strongest association with negative outcomes. The inclusion of numerous longitudinal studies adds further weight to the evidence presented.

### Limitations and Future Directions

One of the main limitations of this review is the considerable heterogeneity among the included studies, especially regarding the assessment methods for exposure and outcomes. This variability prevented the performance of a meta-analysis. The inconsistent reporting of some age groups also hindered the precise definition of the age range of the studied populations. Due to these methodological differences, the findings should be interpreted with caution, as they may have significantly influenced the results. The search strategy used may limit this review, as it employed general terms without specific descriptors for each heavy metal. This approach aimed to verify whether certain metals appeared more frequently in the literature. However, it may have restricted the number of studies identified, and, thus, the results should not be considered a comprehensive overview of the scientific production on the subject. Future research should aim for greater standardization in the methods used to assess exposure and outcomes. Furthermore, studies focused on preadolescent populations are needed to deepen the understanding of the neurotoxic risks associated with heavy metal exposure.

## Figures and Tables

**Figure 1 ijerph-22-01308-f001:**
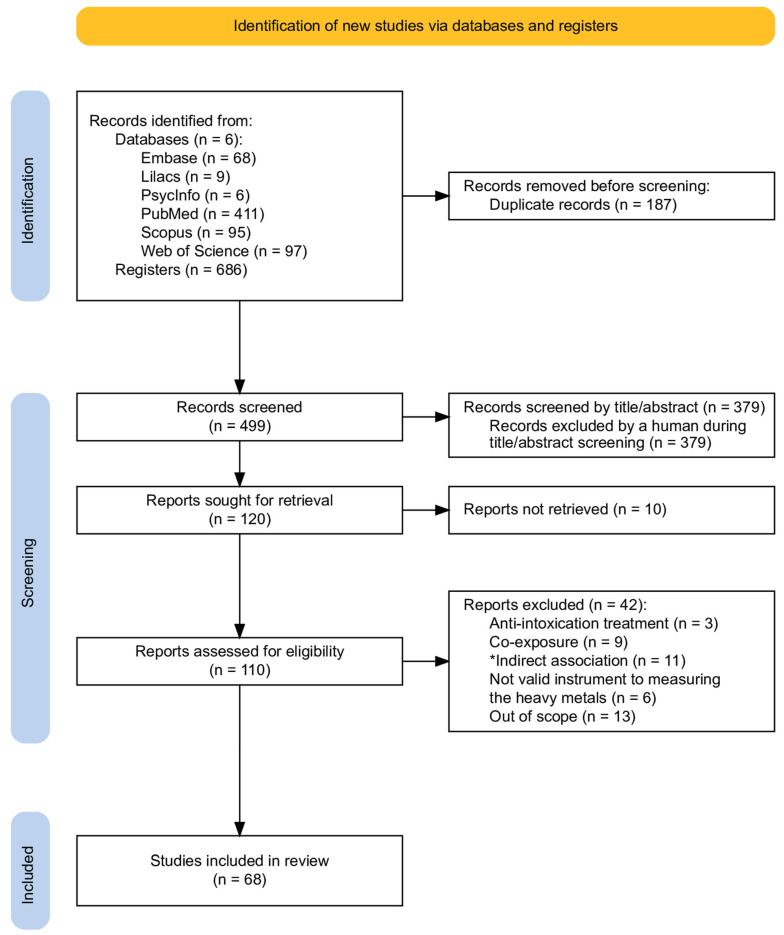
PRISMA Flowchart. * Indirect associations: studies that did not perform a direct association between heavy metal exposure and neurodevelopmental outcomes.

**Figure 2 ijerph-22-01308-f002:**
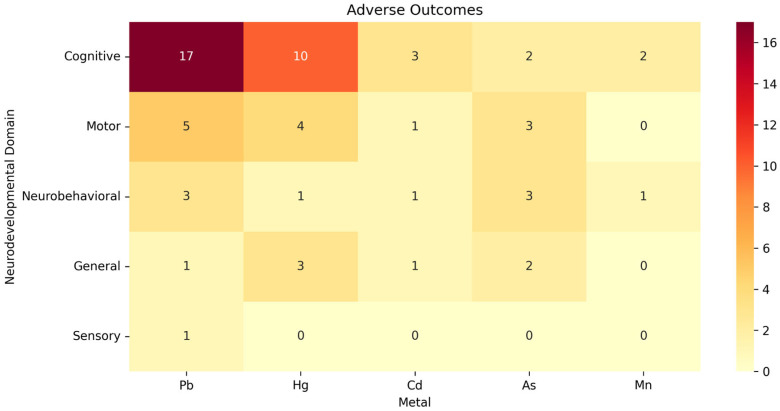
Distribution chart of adverse associations between heavy metals and neurodevelopmental areas by number of studies.

**Figure 3 ijerph-22-01308-f003:**
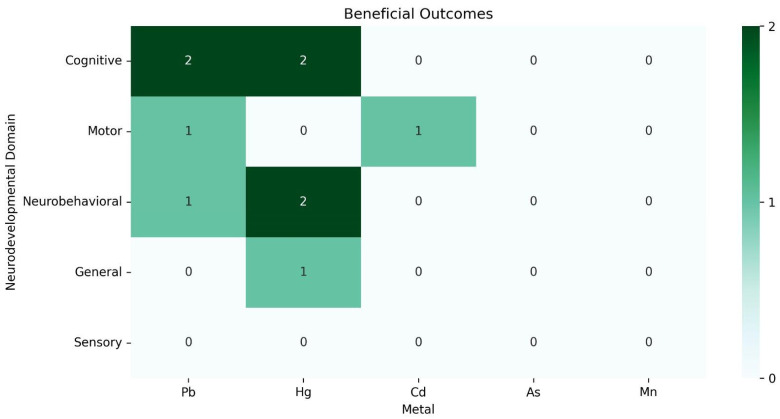
Distribution chart of beneficial associations between heavy metals and neurodevelopmental areas by number of studies.

**Figure 4 ijerph-22-01308-f004:**
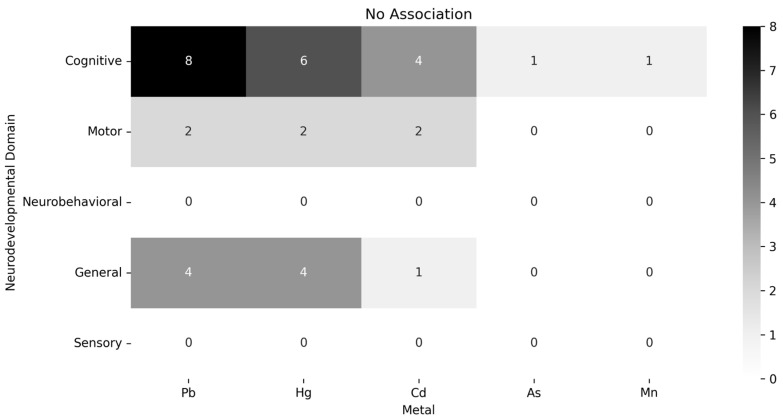
Distribution chart of null associations between heavy metals and neurodevelopmental areas by number of studies.

**Table 1 ijerph-22-01308-t001:** Sample by country.

Country	N Studies	N Sample
Japan	5	181,145
China	12	8485
Spain	5	4479
Norway	2	4375
South Korea	6	2863
Saudi Arabia	3	1970
United Kingdom	1	1558
Mexico	4	1481
Brazil	4	1167
USA	4	1057
India	2	992
Taiwan	5	762
Italy	2	760
Canada	2	727
Seychelles	1	643
Greece	1	575
Bangladesh	1	525
Tanzania	1	439
Poland	1	402
Nepal	3	300
Bolivia	1	246
Malaysia	1	155
Democratic Republic of Congo	1	89
Total	68	215,195

## Data Availability

No new data were created or analyzed in this study. Data sharing is not applicable to this article.

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
