# Peer review of "Neurodevelopmental Outcomes Associated with Early-Life Exposure to Heavy Metals: A Systematic Review"

_ijerph, 2025, doi:10.3390/ijerph22081308_

Round 1

Reviewer 1 Report

Comments and Suggestions for Authors

The authors review heavy metal exposure and neurodevelopmental outcomes in children. Although they had a tough job in summarizing such various literature, and did a great job in reviewing also the quality of the papers included in the review, I have many comments and unfortunately do not think that the manuscript is suitable for publication yet. Many improvements can be made, mainly in the results and discussion section, but also in the rationale and methods.

Introduction: I feel a bit repetition in the introduction, I think it can be more concise. E.g. Heavy metals as a risk for neurodevelopment in line 63 and 77. I do not see the research gap: you mention a lack of broader geographical representation with many studies being from low-and-middle-income countries (in?) Asia (line 110), but I do not see geographical representation in your aim (line 112). Please elaborate more on the novelty/additional value of this review.

Minor comments introduction:

Line 47: The global estimation data refers to a paper from 2018, could more recent data be used for this statement? 

Line 54: This sentence does not end with a dot. 

Line 56-64: Dermal contact is mentioned, however not elaborated on in relation to the infant skin. I think this is a valuable addition, as jewelry, nail polish and personal hygiene/cosmetic products are later mentioned as important exposure sources. 

Line 95-97: This sentence doesn’t read well due to the order or abnormalities listed. 

Methods: I miss some information: please mention when you did the data search. In the flow chart you mention ‘automation tools’, what tools did you use and to what purpose? And what do you regard as neurodevelopment?

Line 114 (Methods): The descriptors do not seem all-encompassing, papers using varieties on the terms (e.g. infant*/juvenile) may be missed. As well as studies that mention just one metal, and therefore not using the term “heavy metals”.  

Minor comments methods:

Paragraph 2.2 (line 133), it reads difficult because it is a sum up of criteria. To increase readability I suggest to change the sentences by adding ‘were excluded’, or other ways, like you did in the final sentence of the paragraph.

Line 147: please change ‘by’ into ‘after’.

Line 158: why 8 or 11? Up to seems to me the maximum and that would be 11 or not?

Results: This section needs improvement. I miss structure, I do not see information about the relationship between heavy metal exposure and neurodevelopment (except from worse vs better outcomes?). A big part of 3.3 is sources of metal exposure, but that is not your aim right? The latter can be discussed, but a different paragraph would be more suitable, this is not a ‘main association’ (you can also think of another title). I also struggle with ‘worse outcomes’ and ‘better outcomes’, it is only a reflection of the number of studies right? There is no information about the size / magnitude of the effect, no information about significance and no information about the quality of the studies. I suggest (and that is usually done) to present a forest plot with ORs and a confidence interval. I understand that you cannot sum up results and do a meta-analysis because the metal is not the same, but there must be a way to make a distinction between the metals.

Figure 1: “Records excluded by a human = 379”, more information would be helpful. Did you record how many papers were reviews for example. This could either be added to the PRISMA flowchart, the text, or a more extensive selection procedure overview could be provided in supplement 1. 

Line 321 It is good to study the quality of the studies, but that needs to be reported in relation to the findings. So, if a study shows ‘better’ neurodevelopmental outcome but is of low quality we interpret that finding different. You can add the scores in Table 4 or visualize it in another way. There are plenty of papers that can be used as inspiration.

Table 3: This table holds too little information, I missed age, gender, number of participants, type of bio sample and neurodevelopmental assessment tool. I suggest to replace Table 3 with Supplementary material 2 (Table 4), as this table does include all of the former listed information (but please use one column for one thing, e.g. author (year) – country - type of study).

Minor comments for the results:

'3.1. Character': should that be characteristics?

Paragraph 3.2 please always use the full name with abbreviation in brackets the first time the abbreviation is mentioned. Some abbreviations, e.g. Mn, were already introduced.

Line 240, delete ‘only’ to ensure objectivity.

Line 241 and further: please add maternal / child to the type of sample: maternal blood, maternal urine or child urine, etc.

Line 264: ‘even those’, too subjective, please delete

Discussion: This needs much improvement as well. Partly that needs to be improved after improvement of the results (you can elaborate then on evidence and quality of the evidence). But also other things are insufficient which concerns both structure and content. For example, in paragraph starting 358 you start with a sentence about neuropsychomotor development, but later on it is about ASD. Also you mention 2 studies with completely different exposures and outcomes: Hg and intelligence tests and Pb and ASD. That does not make sense. And, a sentence like ‘Two other studies found noteworthy results: Abd Wahil et al., (2021) and Al-Saleh et al., (2020).’ does not say anything and should be deleted.

Minor comments discussion:

Line 416-426: This paragraph is mainly a list of results and it repetitively uses ‘name found… name argued: please make the connection between results.

Line 319-321: I don’t understand this sentence. Could it be that the "but" should be removed?  

Line 323: Interesting comment about the possible beneficial effect of simultaneous fish intake. I would appreciate some elaboration on this. This is done a bit later (line 428), consider to mention earlier.

Line 347: which survey? Do you mean review?

Line 428: I like this paragraph, it has one topic, uses both your findings and puts it into perspective with other literature. Would be good to use such a structure in the other parts of the discussion as well.

Supplementary 1: The mentioned combinations for search terms appear incorrect. E.g. when copying the exact PubMed command in PubMed only gives 58 results, while the PRISMA flowchart shows 411.  

Comments on the Quality of English Language

see previous

Author Response

Comments and Suggestions for Authors

The authors review heavy metal exposure and neurodevelopmental outcomes in children. Although they had a tough job in summarizing such various literature, and did a great job in reviewing also the quality of the papers included in the review, I have many comments and unfortunately do not think that the manuscript is suitable for publication yet. Many improvements can be made, mainly in the results and discussion section, but also in the rationale and methods.

We sincerely thank the reviewer for their careful reading, constructive observations, and thoughtful comments. We acknowledge that, despite our efforts to compile a broad and diverse body of literature, there is room for improvement. All comments will be carefully considered in the revised version, and we are committed to enhancing the clarity, coherence, and scientific contribution of our work.

Introduction: I feel a bit repetition in the introduction, I think it can be more concise. E.g. Heavy metals as a risk for neurodevelopment in line 63 and 77. I do not see the research gap: you mention a lack of broader geographical representation with many studies being from low-and-middle-income countries (in?) Asia (line 110), but I do not see geographical representation in your aim (line 112). Please elaborate more on the novelty/additional value of this review.

We appreciate the comment and agree that it is necessary to make the text more concise and to better articulate the research gap, clarifying the added value of this review. We have extensively revised the introduction to address the suggested improvements, which now reads as follows:

“Contamination by heavy metals such as arsenic (As), lead (Pb), cadmium (Cd), and mercury (Hg) poses a serious threat to human health, according to the World Health Organization [1]. These metals accumulate in the food chain, water, and dust, becoming sources of exposure through ingestion, inhalation, or dermal contact [2]. Although dermal absorption of heavy metals is generally low [3], it is especially relevant for children, whose skin is thinner, more permeable, and less resistant to irritants and infections [4]. Children are particularly vulnerable because they ingest and absorb more Pb per unit of body weight than adults [5], and they often engage in risk behaviors such as putting objects or their hands into their mouths [2]. Since early childhood is a critical period for neurological development, exposure to heavy metals may affect the immature nervous system and contribute to neurodevelopmental disorders [6, 7].

Child neurodevelopment is broadly assessed across cognitive, language, motor, and social domains. Delays in any of these areas indicate developmental progress below what is expected for a given age group [8]. Neurobehavioral development, which involves brain maturation processes responsible for regulating behavior, cognition, and emotions, is essential for academic performance and psychological well-being [9]. It is estimated that neurobehavioral disorders affect between 10% and 15% of children worldwide, with a significant global rise in the prevalence of autism spectrum disorder (ASD) and attention-deficit/hyperactivity disorder (ADHD) [10, 11]. The proper maturation of the central nervous system requires adequate levels of essential elements such as copper (Cu), zinc (Zn), iron (Fe), and selenium (Se) [12]. In contrast, transition metals like manganese (Mn) and heavy metals are toxic even at low concentrations and can impair neurodevelopment beginning in the prenatal period [13].

The literature highlights exposure to heavy metals as a contributing factor in the etiology of neurodevelopmental disorders [14]. However, the magnitude of this association has not yet been fully established, as determining causality and linking specific agents remains challenging [15]. According to comprehensive estimates from the United Nations Children’s Fund (UNICEF), 10.1% of individuals aged 0 to 17 years worldwide have moderate to severe disabilities, with children aged 0 to 4 years accounting for 12.2% of this total [16] (Olusanya et al., 2022). Additionally, it is estimated that 200 million children under the age of five in developing countries are not reaching their full neurodevelopmental potential [17].

Pb is one of the most widely studied neurotoxins and is associated with intellectual and motor deficits, behavioral changes, and mood regulation dysfunctions due to its interference with gene expression during critical periods of brain plasticity [18, 19]. Its occupational sources are not yet clearly defined, and most studies focus on specific regions [20]. Areas near mining operations pose elevated risks, with blood Pb levels higher than those in other mining zones, and residues from these activities may continue to contaminate the environment and food chain even after mining has ceased [21, 22, 23]. Other sources include both formal and informal activities such as battery manufacturing, radiator repair, artisanal production, and the use of lead in fishing weights, jewelry, decorative items, and glazes, potentially affecting women and children [20]. Additionally, personal hygiene products and cosmetics may contain Pb, As, and Hg [24, 25].

Exposure to heavy metals during pregnancy has been associated with various neurodevelopmental outcomes. Pb has been linked to an increased incidence of ASD [26], while Cd appears to negatively affect overall neurodevelopment and has also been associated with epilepsy [13]. Additionally, exposure to Hg and As has been connected to structural alterations in the frontal and subcortical brain regions, as well as to dysregulation of neurotransmission [7].

Given the multiple environmental sources of heavy metals and the associated risks of exposure during childhood, several systematic reviews have investigated specific relationships between individual metals and outcomes such as ADHD [27] (Gu et al., 2024), ASD [28, 29], cognitive performance [30], and general health problems [31, 32]. While these reviews reflect growing concern about the impact of heavy metals on child development, an updated and comprehensive synthesis is needed to organize scattered evidence, integrate different types of exposure, and clarify associations across various neurodevelopmental domains, including motor, cognitive, sensory, social, and behavioral. This review aims to fill this gap by systematically gathering evidence on the association between heavy metal exposure and impairments or, in some cases, advantages in neurodevelopmental performance domains.”

Minor comments introduction:

Line 47: The global estimation data refers to a paper from 2018, could more recent data be used for this statement? 

We thank you and agree with the observation. We have revised the introduction to include a more up-to-date reference; the text now reads as follows:

“According to comprehensive estimates from the United Nations Children’s Fund (UNICEF), 10.1% of individuals aged 0 to 17 years worldwide have moderate to severe disabilities, with children aged 0 to 4 years accounting for 12.2% of this total [16] (Olusanya et al., 2022). Additionally, it is estimated that 200 million children under the age of five in developing countries are not reaching their full neurodevelopmental potential [17].”

Line 54: This sentence does not end with a dot. 

We appreciate the comment. We replaced the semicolon with a period to improve the clarity and flow of the sentence.

Line 56-64: Dermal contact is mentioned, however not elaborated on in relation to the infant skin. I think this is a valuable addition, as jewelry, nail polish and personal hygiene/cosmetic products are later mentioned as important exposure sources. 

We thank the reviewer for the relevant comment regarding dermal exposure and its significance for children's skin. We have revised the paragraph to include a more detailed discussion of this exposure pathway: “Although dermal absorption of heavy metals is generally low [3], it is especially relevant for children, whose skin is thinner, more permeable, and less resistant to irritants and infections [4].”

Line 95-97: This sentence doesn’t read well due to the order or abnormalities listed. 

We thank the reviewer for the comment. We have revised this passage to improve its clarity and flow: “Exposure to heavy metals during pregnancy has been associated with various neurodevelopmental outcomes. Pb has been linked to an increased incidence of ASD [26], while Cd appears to negatively affect overall neurodevelopment and has also been associated with epilepsy [13]. Additionally, exposure to Hg and As has been connected to structural alterations in the frontal and subcortical brain regions, as well as to dysregulation of neurotransmission [7].”

Methods: I miss some information: please mention when you did the data search.

We thank the reviewer for the comment. Regarding the search date, this information was already included in the Methods section of the original manuscript. The search was conducted on 02/15/2025.

In the flow chart you mention ‘automation tools’, what tools did you use and to what purpose?

Automation tools' refer to non-human tools used to exclude ineligible articles during systematic reviews, such as other reviews, and this option appears by default in the new PRISMA flowcharts, reflecting advances in screening technologies. However, since our entire selection process was conducted manually, we will remove this item from the flowchart.

And what do you regard as neurodevelopment?

We also revised the Methods section to include a definition of neurodevelopment as considered in this review. The revised text now reads: “For the purposes of this systematic review, neurodevelopment refers to the structural and functional processes involved in the formation of the central nervous system from gestation, which result in the development of sensory, motor, cognitive, language, social, and behavioral functions in children after birth [34, 35].”

Line 114 (Methods): The descriptors do not seem all-encompassing, papers using varieties on the terms (e.g. infant*/juvenile) may be missed. As well as studies that mention just one metal, and therefore not using the term “heavy metals”.  

We appreciate the important observation. We agree that the descriptors might seem restrictive; however, preliminary searches were conducted to test the retrieval results of the descriptor combinations. Including other population descriptors was not effective in significantly increasing the number of articles specific to the childhood phase. Moreover, the use of these descriptors allowed the inclusion of studies that addressed only a single heavy metal. 

Minor comments methods:

Paragraph 2.2 (line 133), it reads difficult because it is a sum up of criteria. To increase readability I suggest to change the sentences by adding ‘were excluded’, or other ways, like you did in the final sentence of the paragraph.

We appreciate the comment. We have made the necessary changes, as requested by the reviewers, to clarify the information. The text now reads as follows: “Studies on neurodegenerative diseases in childhood were excluded. Studies on neurodevelopmental disorders were excluded if they did not report diagnoses based on validated instruments. According to the Diagnostic and Statistical Manual of Mental Disorders, Fifth Edition, Text Revision (DSM-5-TR), neurodevelopmental disorders begin in the early stages of life and generally follow a stable course, as seen in intellectual disability or ASD [36] American Psychiatric Association, 2022). In contrast, pediatric neurodegenerative diseases have a progressive and irreversible course, characterized by the loss of previously acquired neurological skills, as observed in leukodystrophies and lysosomal storage diseases [37] Mastrangelo et al., 2019). By “validated instrument” for the diagnosis or assessment of neurodevelopment, we refer to standardized tools with adequate psychometric properties (such as validity and reliability) that are widely accepted in the scientific literature, for example, the Bayley Scales of Infant and Toddler Development (Bayley-III or Bayley-4), used to assess cognitive, motor, and language development in children aged 1 to 42 months. This criterion aims to ensure the inclusion of only studies employing rigorous assessments through standardized instruments, excluding those based solely on expert opinion or informal reports from parents and caregivers.

Studies in which exposure to heavy metals was not demonstrated by validated individual biomonitoring measures were also excluded. Investigations of co-exposures that did not allow extraction of specific data on heavy metals were excluded. This criterion aims to minimize potential confounding factors by excluding studies that assess other toxic elements besides heavy metals, as well as nutrients that can be replenished through supplementation, such as iron. Studies focused on trace element supplementation or iron deficiency anemia, as well as markers of iron metabolism such as transferrin and ferritin, were excluded. Reviews of all types, case reports, case series, gray literature, letters, editorials, and scientific meeting documents were also excluded.”

Line 147: please change ‘by’ into ‘after’.

We appreciate the opportunity to improve our work. The change was made.

Line 158: why 8 or 11? Up to seems to me the maximum and that would be 11 or not?

We appreciate the feedback. The total score ranges from 8 to 11 depending on the number of items in the Joanna Briggs Institute (JBI) Critical Appraisal Tool, which varies according to the type of study analyzed, 8 items for cross-sectional studies and 11 for longitudinal studies. Accordingly, we have added this information to the text to clarify this methodological process.

Results: This section needs improvement. I miss structure, I do not see information about the relationship between heavy metal exposure and neurodevelopment (except from worse vs better outcomes?). A big part of 3.3 is sources of metal exposure, but that is not your aim right? The latter can be discussed, but a different paragraph would be more suitable, this is not a ‘main association’ (you can also think of another title). I also struggle with ‘worse outcomes’ and ‘better outcomes’, it is only a reflection of the number of studies right? There is no information about the size / magnitude of the effect, no information about significance and no information about the quality of the studies. I suggest (and that is usually done) to present a forest plot with ORs and a confidence interval. I understand that you cannot sum up results and do a meta-analysis because the metal is not the same, but there must be a way to make a distinction between the metals.

We thank you for the valuable contribution. We agree and made changes where possible to accommodate the results in a straight-forward, clear and efficient way. After the changes, the text is now presented like so: “3.3. Main Associations

The available data could not be meta-analyzed due to the heterogeneity of exposure assessment methodologies, exposure thresholds, and outcomes assessed, which prevented direct comparison. Therefore, we performed a data synthesis for the consistently investigated heavy metals: Pb, Hg, As, Cd, and Mn. Information on each of the 68 included studies is available in Table 3, and more details are available in supplementary material 2 - Table 4. The associations between metal exposure and neurodevelopmental outcomes are graphically represented in Figures 2, 3, and 4. Associations indicating impaired neurodevelopment consistently outnumbered those that found a positive effect or even those in which no association was found. In both cross-sectional and longitudinal studies, studies of above-average quality were the majority. Most studies investigated the influence of heavy metal exposure on the neurodevelopment of children during the intrauterine period. Therefore, the evidence generated in this review is more robust regarding prenatal exposure to heavy metals.

Of the 68 included studies, 29 (42.6%) presented sufficient data to estimate the effect size of the associations between heavy metal exposure and neurodevelopmental outcomes. The majority (79.3%) indicated a small effect, while moderate and large effects were observed in 13.8% and 17.2% of the studies, respectively. Some studies assessed more than one outcome. Among these 29 studies, 55.2% were classified as good quality, 41.4% as fair quality, and 3.4% as low quality. The results are organized and described according to the associated heavy metal and the observed effect size. Statistical details are provided in Table 3.

3.3.1 Associations with small effect sizes

Mercury (Hg) and Methylmercury (MeHg)

Several neurodevelopmental outcomes have been associated with Hg and MeHg. Two fair-quality studies reported associations between Hg exposure and improved global and motor neurodevelopmental scores [41, 42]. Conversely, six studies identified associations between Hg or MeHg exposure and impairments in different domains, including social skills [43], behavioral problems and cognition [44, 45], mental development [42], and risk of global neurodevelopmental delay [46, 47]. The studies by Myers et al. [43], Marques et al. [41, 42], Freire et al. [45] and Nyanza et al. [46] were assessed as of fair quality, while those by Al-Saleh et al. [44, 47] were classified as of good quality.

Lead (Pb)

Negative impacts on several neurodevelopmental domains have been associated with Pb in twelve studies. Pb exposure was related to lower cognitive performance [48, 49, 50, 51, 52, 53], gross motor function [54], higher risk of ADHD [56], sensory processing dysfunction [56], lower mental development [58], lower psychomotor development [59], and lower global neurodevelopment [60]. The studies by Guo et al. [50], Liu et al. [52], and Park et al. [55] were assessed as of fair quality, while the others were classified as of good quality.

 Arsenic (As)

Good to fair quality studies have shown that As exposure was associated with worsening cognition, social skills, motor development, increased risk of ASD, and global neurodevelopmental performance [60, 61, 62, 46].

Cadmium (Cd)

Small effect size associations for Cd indicated increased risk of ADHD and ASD, as well as worsening motor function. These effects were observed in good to fair quality studies [61, 40, 60].

Manganese (Mn)

Exposure to Mn exposure has been associated with increased risk of ASD and poorer cognitive performance in good quality studies [61, 62].

3.2.2 Associations with medium effect sizes

Unfavorable associations were observed with neurodevelopmental outcomes, particularly with Hg and Pb. Among good-quality studies, Kim et al. [63] found that Hg exposure was associated with poorer mental development. Among fair-quality studies, Al-Saleh et al. [65] reported that higher Hg levels were associated with lower cognitive function scores and a higher risk of visuomotor integration problems. A fair-quality study found worse language performance among children exposed to Hg [46]. Finally, a low-quality study found an association between Pb exposure and lower cognition scores [66].

3.3.3 Associations with large effect sizes

Adverse effects with large effect sizes were identified for Pb, Cd, and Hg, particularly affecting cognition, behavior, and global development. Among good-quality studies, Nie et al. [67] reported that Pb exposure was associated with worse cognitive and behavioral performance. Also in this group, Kim et al. [51] observed that Hg was associated with worse psychomotor performance, and Kao et al. [67] found that Cd exposure was associated with worse cognitive outcomes. Among fair-quality studies, Halabicky et al. [69] found an association between Pb and lower cognitive performance, while Nyanza et al. [46] reported that Cd was related to impairments in global neurodevelopment.”

Figure 1: “Records excluded by a human = 379”, more information would be helpful. Did you record how many papers were reviews for example. This could either be added to the PRISMA flowchart, the text, or a more extensive selection procedure overview could be provided in supplement 1. 

We thank the reviewer for this pertinent comment. We agree that providing more detailed information on records excluded during the screening phase could enhance the transparency of the review process. We acknowledge the value of this suggestion and will be attentive to improving our methodology in future reviews to allow reporting of exclusion reasons from the initial stages. In this review, we followed the PRISMA 2020 guidelines and detailed exclusion reasons starting from the eligibility phase (full-text reading), as recommended. These data are already included in the flowchart. We've edited the text in the method section to make this information clearer: “Articles that were not included after full reading had the reason for exclusion stated in the PRISMA flowchart.”

Line 321 It is good to study the quality of the studies, but that needs to be reported in relation to the findings. So, if a study shows ‘better’ neurodevelopmental outcome but is of low quality we interpret that finding different. You can add the scores in Table 4 or visualize it in another way. There are plenty of papers that can be used as inspiration.

We appreciate this observation. We agree that assessing the quality of the studies is a very important aspect of the evidence generated by the review. We were challenged by the large amount of data produced and the limited space available in the article to communicate everything effectively. Encouraged by your comment, we have relocated some data and added the methodological quality of the studies to Table 3.

Table 3: This table holds too little information, I missed age, gender, number of participants, type of bio sample and neurodevelopmental assessment tool. I suggest to replace Table 3 with Supplementary material 2 (Table 4), as this table does include all of the former listed information (but please use one column for one thing, e.g. author (year) – country - type of study).

We appreciate the observation. We have relocated some review data to provide more results in the main table, Table 3. However, it was not possible to allocate a separate column for each item due to the space allowed by the formatting. The additional data are provided in the supplementary material, Table 4.

Minor comments for the results:

'3.1. Character': should that be characteristics?

We appreciate the opportunity to improve our work and, accordingly, have amended the above term as requested.

Paragraph 3.2 please always use the full name with abbreviation in brackets the first time the abbreviation is mentioned. Some abbreviations, e.g. Mn, were already introduced.

We appreciate the feedback. All terms have been reviewed and, when previously mentioned, have been replaced by their abbreviations.

Line 240, delete ‘only’ to ensure objectivity.

We appreciate the observation. The above term has been removed as requested.

Line 241 and further: please add maternal / child to the type of sample: maternal blood, maternal urine or child urine, etc.

We appreciate the opportunity to improve our work. We have edited the mentioned text to accommodate the suggested improvements, which now reads as follows:

There was also considerable variation in the type of biological samples analyzed, with some studies using samples from the mother, the child, or both. Regarding maternal samples, 32 studies analyzed blood (47,06%), 10 analyzed urine (14.71%), 7 analyzed hair (10.29%), and 4 analyzed breast milk (5,88%). For child-derived samples, blood was analyzed in 37  studies (54.41%), hair in 10 (14.71%), urine in 7 (10.29%), and nails in only 2 (2.94%). Among all sample types, blood was the most frequently analyzed material, being exclusively examined in 43 studies (63.23%). Urine was used exclusively in 9 studies (13.23%), while hair or a combination of hair and nails was analyzed in 8 studies (11.76%).”

Line 264: ‘even those’, too subjective, please delete

We appreciate the comment. The above term has been removed as requested.

Discussion: This needs much improvement as well. Partly that needs to be improved after improvement of the results (you can elaborate then on evidence and quality of the evidence). But also other things are insufficient which concerns both structure and content. For example, in paragraph starting 358 you start with a sentence about neuropsychomotor development, but later on it is about ASD. Also you mention 2 studies with completely different exposures and outcomes: Hg and intelligence tests and Pb and ASD. That does not make sense. And, a sentence like ‘Two other studies found noteworthy results: Abd Wahil et al., (2021) and Al-Saleh et al., (2020).’ does not say anything and should be deleted.

We greatly appreciate the constructive criticism. We edited the Discussion section, now it reads as follows:

“The findings of this systematic review indicate that exposure to heavy metals is predominantly associated with impairments in child neurodevelopment. Most studies report adverse effects, although there is variability in the magnitude of these effects and the domains affected. Among the metals analyzed, Pb presents the most consistent evidence, followed by Hg, As, Cd, and Mn. Most associations involve deficits in the cognitive domain, followed by behavioral and motor outcomes, suggesting that different neurobiological pathways may be involved. A recurring theme in high-quality studies is the greater vulnerability of the nervous system during pregnancy, especially in the first trimester. This period shows more robust associations, supporting the hypothesis of critical windows in fetal development [106, 107]. The plasticity and immaturity of the developing brain make it particularly susceptible to adverse environmental influences, including persistent and cumulative chemical exposures [108].

Previous studies have identified Pb, Hg, As, and Cd as the main metals associated with neurodevelopmental impairments, with Pb’s adverse effects being particularly consistent [109]. Recent evidence further supports these findings, especially regarding cognitive outcomes [15], and studies focusing on lead report greater impairments in childhood neurobehavior [110]. The meta-analysis by Rosenauer et al. [111] found that high blood Pb levels increase the risk of ADHD, linked to neuronal alterations such as synapse reduction and dysfunctions in neurotransmission and calcium signaling. These changes affect key brain structures involved in attention and behavioral control and involve dysfunctions in the dopaminergic and noradrenergic systems. Inhibitory control, a cognitive function essential for regulating behavior, is compromised in ADHD and was also impaired in participants in the study by Desye et al. [110], reinforcing the strong association between Pb exposure and cognitive deficits identified in our review.

The neurotoxic effects of heavy metals, however, do not occur in isolation. Several biological and physiological factors can modulate their toxicity depending on individual characteristics. One such modulator is the gut microbiota, whose composition influences the bioavailability of metals and can alter their absorption and accumulation in the body [112]. Experimental studies have shown that mice on dietary alterations, such as gluten-free diets, accumulate more metals in their tissues [113], highlighting the microbiome’s role. Because this system also participates in neural maturation, its dysfunction can impact neurodevelopment through the gut-brain axis [114]. The interaction between metals and microbiota is bidirectional: while metals affect microbial composition, the microbiota can either mitigate or intensify the toxic effects of these elements [112]. Evidence also suggests that alterations in the microbiome relate to behavioral manifestations, as observed in mice that received microbiota from individuals with ASD [115]. The findings of this review reinforce the association of ASD and ADHD with elevated levels of metals such as Pb, Cd, As, and Mn [61, 19, 79, 55], indicating that shared neurochemical and developmental pathways may be involved in the etiology of these disorders [61].

Although many studies highlight the negative effects of heavy metals on neurodevelopment, ten articles in this review reported some positive effects, particularly related to Hg [83, 73, 103, 66, 41, 43]. A common explanation for these findings is the higher fish consumption among the populations studied. While heavy metals alone impair neurocognition, their presence alongside seafood intake may be linked to cognitive benefits [116, 117]. The review by Hibbeln et al. [117] suggests that fish consumption can mitigate the harmful effects of heavy metals. Supporting this, Lyall et al. [118] found that prenatal fish consumption may reduce the likelihood of ASD diagnoses, whereas low consumption was associated with increased risk [119]. However, this latter study acknowledges that these effects cannot be attributed solely to nutrients.

A recent review on seafood consumption and child neurodevelopment found that children aged 10 months to 15 years who consumed fish exhibited better cognitive outcomes [120]. However, in our review, Kou et al. [77] identified negative effects of combined exposure to Cd, Hg, and Pb on language development, possibly exacerbated by fish consumption. It is widely accepted that nutrition from the prenatal period through the first two years of life is critical for neurodevelopment [121]. Deficiencies in essential nutrients may link Pb exposure to neurodevelopmental deficits independently of fish consumption, leading to lasting health impacts [121].

Although most studies included in this review identified associations between heavy metal exposure and impaired child neurodevelopment, effect sizes were established in less than half of them. Furthermore, most of these associations showed small effect sizes, which limits the robustness of the conclusions, as small effects may reflect fragile associations susceptible to bias and methodological variations. However, a considerable number of studies reported moderate to large effect sizes, indicating that in some cases the impacts may be more pronounced. This variation suggests that factors such as exposure levels and duration, environmental interactions, and individual characteristics can influence the magnitude of neurotoxic effects. Moreover, although small effect sizes are often considered less clinically relevant, they can have significant public health implications due to the widespread and cumulative exposure to these substances.

Another important aspect concerns the methodological quality of the studies. Most studies included in this review had a longitudinal design, which strengthens the findings by enabling analysis of temporal relationships between exposure and neurodevelopmental outcomes. Among studies with data available to estimate effect sizes, the majority were classified as good quality, including many that found small associations. This supports the idea that smaller effects are not necessarily due to bias or methodological flaws, but may reflect genuine, subtle effects of these exposures. Conversely, studies reporting moderate and large effects ranged from good to fair quality, indicating that a larger effect size alone does not guarantee greater reliability. Therefore, combining effect size assessment with study quality is essential for properly interpreting the available evidence. This underscores the need for future research investigating mechanisms, specific exposures, and potential moderators to clarify the clinical relevance of these findings.”

Minor comments discussion:

Line 416-426: This paragraph is mainly a list of results and it repetitively uses ‘name found… name argued: please make the connection between results.

Thank you for the observation. We agree, therefore we have edited the text as a whole in the discussion, modifying this passage in the process.

Line 319-321: I don’t understand this sentence. Could it be that the "but" should be removed?  

We appreciate the insightful comment. The sentence has been revised to improve cohesion and clarity.

Line 323: Interesting comment about the possible beneficial effect of simultaneous fish intake. I would appreciate some elaboration on this. This is done a bit later (line 428), consider to mention earlier.

Thank you for the feedback! We have extensively revised the discussion section to address the reviewers’ comments, and this information has been moved to an earlier part of the section.

Line 347: which survey? Do you mean review?

We appreciate the insightful observation. The term 'survey' does indeed refer to our review, and therefore we will replace the term for better clarity.

Line 428: I like this paragraph, it has one topic, uses both your findings and puts it into perspective with other literature. Would be good to use such a structure in the other parts of the discussion as well.

Thank you for the positive comment. We have revised the discussion section to incorporate all of your suggested improvements.

Supplementary 1: The mentioned combinations for search terms appear incorrect. E.g. when copying the exact PubMed command in PubMed only gives 58 results, while the PRISMA flowchart shows 411. 

We appreciate the insightful observation. The syntax error has been corrected and the search now returns a number of articles equivalent to that obtained at the beginning of the systematic review.

Comments on the Quality of English Language

see previous

The manuscript has been carefully revised for readability.

We thank the reviewer for the opportunity to improve our work and for the valuable contributions provided throughout the revision process.

Reviewer 2 Report

Comments and Suggestions for Authors

The number of studies identified for inclusion (68 studies) seems low, considering that this systematic review has a very broad scope (any studies looking at heavy metals and neurodevelopment). The only search terms were "heavy metals" and "neurodevelopment". I suspect that more studies would have been identified if the search terms had also included the names of specific heavy metals (lead, cadmium, arsenic, etc) and specific neurodevelopmental disorders. 

In the methods section, the inclusion and exclusion criteria could be better described:

  • Can you give examples of what you are considering to be a neurodegenerative disease (excluded) versus a neurodevelopmental disorder (included)?
  • Can you give examples of what would be considered a "validated instrument" for diagnosis?
  • Regarding co-exposures, there are many epidemiology studies that look at exposure to suites of different metals. Would these be included as long as the metals were analyzed individually?

This PROSPERO number leads to the protocol for a systematic review of heavy metals and semen quality and sex hormone levels (not neurodevelopment). Please double check the number.

The study evaluation results are currently shown in supplemental Tables 5 and 6. I suggest adding the overall study quality rating (Good/Fair/Poor) to Table 3 of the main text as a column alongside the study findings, since it would aid in the interpretation. I think it would also be preferable to change supplemental tables 5 and 6 to Excel files so that they could be easily filtered and searched (as opposed to static Word tables).

Please also consider listing the risk of bias questions in your supplemental materials; you currently just say Q1, Q2, etc without any explanation for what the questions are.

In Figures 2 and 3, consider changing the titles “Worse Outcomes” and “Better Outcomes” to “Adverse Outcomes” and “Beneficial Outcomes”.

In Figure 3, the scale should be revised so that it only shows whole numbers (0, 1, 2) rather than showing decimals.  

Author Response

Comments and Suggestions for Authors

The number of studies identified for inclusion (68 studies) seems low, considering that this systematic review has a very broad scope (any studies looking at heavy metals and neurodevelopment). The only search terms were "heavy metals" and "neurodevelopment". I suspect that more studies would have been identified if the search terms had also included the names of specific heavy metals (lead, cadmium, arsenic, etc) and specific neurodevelopmental disorders. 

We appreciate the reviewer’s comment and understand the concern regarding the number of studies included. The decision to use more general search terms was intentional and aligned with the research question, which aimed to broadly map the literature to assess whether certain heavy metals would appear more frequently even without explicitly targeting them in the search strategy. This approach allowed us to identify emerging patterns in the literature without introducing an a priori bias toward specific substances or conditions. Furthermore, preliminary tests including specific metal names and clinical outcomes led to a substantial increase in the heterogeneity of the retrieved studies, both methodologically and in terms of populations studied.

In the methods section, the inclusion and exclusion criteria could be better described:

Can you give examples of what you are considering to be a neurodegenerative disease (excluded) versus a neurodevelopmental disorder (included)?

We appreciate the comment. The key difference is that one refers to the loss of acquired skills, while the other reflects the absence of typical development. We have revised the Methods section to clarify this distinction, which now reads as follows:

“According to the Diagnostic and Statistical Manual of Mental Disorders, Fifth Edition, Text Revision (DSM-5-TR), neurodevelopmental disorders begin in the early stages of life and generally follow a stable course, as seen in intellectual disability or ASD [36]. In contrast, pediatric neurodegenerative diseases have a progressive and irreversible course, characterized by the loss of previously acquired neurological skills, as observed in leukodystrophies and lysosomal storage diseases [37].”

Can you give examples of what would be considered a "validated instrument" for diagnosis?

We appreciate the comment. We have included an explanation in the Methods section for greater clarity, as follows:

“By ‘validated instrument’ for the diagnosis or assessment of neurodevelopment, we refer to standardized tools with adequate psychometric properties (such as validity and reliability) that are widely accepted in the scientific literature, for example, the Bayley Scales of Infant and Toddler Development (Bayley-III or Bayley-4), used to assess cognitive, motor, and language development in children aged 1 to 42 months. This criterion aims to ensure the inclusion of only studies employing rigorous assessments through standardized instruments, excluding those based solely on expert opinion or informal reports from parents and caregivers.”

Regarding co-exposures, there are many epidemiology studies that look at exposure to suites of different metals. Would these be included as long as the metals were analyzed individually?

Thank you for the comment. Yes, we applied this criterion to avoid including studies that jointly assessed heavy metals and other toxic elements, which could introduce confounding factors into the results. We have revised the Methods section to clarify this, as follows:

“This criterion aims to minimize potential confounding factors by excluding studies that assess other toxic elements besides heavy metals, as well as nutrients that can be replenished through supplementation, such as iron.”

This PROSPERO number leads to the protocol for a systematic review of heavy metals and semen quality and sex hormone levels (not neurodevelopment). Please double check the number.

We appreciate the insightful observation. The registered number is correct, and the error originated from the PROSPERO platform itself. Fortunately, we contacted their support team, which promptly corrected the inconsistency. The registration number is now correctly displayed for our systematic review.

The study evaluation results are currently shown in supplemental Tables 5 and 6. I suggest adding the overall study quality rating (Good/Fair/Poor) to Table 3 of the main text as a column alongside the study findings, since it would aid in the interpretation. I think it would also be preferable to change supplemental tables 5 and 6 to Excel files so that they could be easily filtered and searched (as opposed to static Word tables).

We appreciate the suggestions. We have added the quality assessment data from supplementary Tables 5 and 6 to Table 3 in the main text and accepted the recommendation to change the format of Tables 5 and 6 from Word to Excel.

Please also consider listing the risk of bias questions in your supplemental materials; you currently just say Q1, Q2, etc without any explanation for what the questions are.

We are grateful for your suggestion and have added an additional supplementary document containing the items related to the risk of bias assessment.

In Figures 2 and 3, consider changing the titles “Worse Outcomes” and “Better Outcomes” to “Adverse Outcomes” and “Beneficial Outcomes”.

We thank you for the suggestion, as the change in words are very much needed in order to not appear insensitive, considering the topic of the research.

In Figure 3, the scale should be revised so that it only shows whole numbers (0, 1, 2) rather than showing decimals.  

We appreciate and thank you for your sharp attention to the details in our manuscript, and fixed the scale numbers. Since the highest number is 2 the scale numbers now repeat, but it does, in fact, seem more logical to only display whole numbers.

We thank the reviewer for the opportunity to improve our work and for the valuable contributions provided throughout the revision process.

Reviewer 3 Report

Comments and Suggestions for Authors

After reviewing the paper, I found it very interesting to see a work that compiles so many papers to know the “state of the art” in such an interesting and topical subject, which is of great concern to the population, such as the relationship between exposure to heavy metals and the appearance of diseases in the population, especially in children, who are so vulnerable.

I believe that it serves as a guide and that it is very interesting and has been done with an adequate methodology for the bibliographic search and screening. It reflects the great concern in the scientific world about these issues and the gaps in knowledge that still need to be investigated.

However, we must be very cautious and, as the same article comments, there is a lot of variability and in many cases neither the methodologies for determining heavy metals, nor the type of biological sample taken, nor many other factors that are particular to each of the studies are comparable. In this sense, this article should be taken as a compilation of how this type of studies, which are of great concern to the general population because they affect the health of children, are being carried out worldwide. They are not comparable works, in most cases.

I see it as suitable for publication taking into account these premises.

Regarding the format, the bibliography is not adapted to the mdpi format in which citations should be numbered in the order of appearance in the text and in the final list should also be listed in order of appearance and not alphabetically.

Author Response

Comments and Suggestions for Authors

After reviewing the paper, I found it very interesting to see a work that compiles so many papers to know the “state of the art” in such an interesting and topical subject, which is of great concern to the population, such as the relationship between exposure to heavy metals and the appearance of diseases in the population, especially in children, who are so vulnerable.

I believe that it serves as a guide and that it is very interesting and has been done with an adequate methodology for the bibliographic search and screening. It reflects the great concern in the scientific world about these issues and the gaps in knowledge that still need to be investigated.

However, we must be very cautious and, as the same article comments, there is a lot of variability and in many cases neither the methodologies for determining heavy metals, nor the type of biological sample taken, nor many other factors that are particular to each of the studies are comparable. In this sense, this article should be taken as a compilation of how this type of studies, which are of great concern to the general population because they affect the health of children, are being carried out worldwide. They are not comparable works, in most cases.

I see it as suitable for publication taking into account these premises.

Regarding the format, the bibliography is not adapted to the mdpi format in which citations should be numbered in the order of appearance in the text and in the final list should also be listed in order of appearance and not alphabetically.

We sincerely thank you for your insightful remarks and for acknowledging the relevance and timeliness of the topic addressed. We share your concern regarding the methodological variability of the included studies and emphasized it in the limitations section of our review to appropriately guide the interpretation of the findings:

One of the main limitations of this review is the considerable heterogeneity among the included studies, particularly in how exposure and outcomes were assessed. This variability made it impossible to perform a meta-analysis. Inconsistent reporting of age groups also made it difficult to accurately define the age range of the study populations. Due to these methodological differences, the findings should be interpreted with caution, as they may have significantly influenced the results. Future research should aim for greater standardization in both exposure and outcome assessment methods. Moreover, studies focusing on pre-adolescent populations are needed to deepen our understanding of the neurotoxic risks posed by heavy metal exposure.”

Regarding the formatting, we will adjust all citations and references according to MDPI guidelines, with numbering based on their order of appearance in the text and corresponding organization in the final reference list.

We thank you once again for your careful reading and constructive contributions.

Round 2

Reviewer 2 Report

Comments and Suggestions for Authors

I had commented on the original submission that the number of studies identified in the review was rather small, likely due to the lack of specific search terms. The authors responded: “The decision to use more general search terms was intentional and aligned with the research question, which aimed to broadly map the literature to assess whether certain heavy metals would appear more frequently even without explicitly targeting them in the search strategy.” This is fine, but the authors should acknowledge this as a limitation of the review. The identified studies should not be considered a comprehensive overview of the subject.

The revised manuscript talks about the results in terms of small, medium, and large effect sizes, which is not something that was in the original draft. The thresholds for small, medium, and large are unclear and need to be defined by the authors. I think it could be useful to talk about effect sizes, but I do not think that current organization of the text makes much sense; e.g., lead and mercury appear in all three sections (small, medium, and large). Consider reorganizing the discussion of effect sizes into a short paragraph or maybe adding a figure that summarizes the effect size findings for different metals.

In Figure 1, consider changing the text in the first screening level to something like “Records screened by title/abstract” and “Records excluded by a human during title/abstract screening”.

At the bottom of Figure 1, what does “Indirect association” refer to? Please specify this somewhere.

The methods section talks about study quality ratings as “high, moderate, low”, whereas the results talks about “good, fair, poor” or sometimes “good, fair, low”. Please use consistent terminology.

Line 409: “However, in our review, Kou et al. [77] identified negative effects of combined exposure to Cd, Hg, and Pb on language development, possibly exacerbated by fish consumption.”  Wording is unclear; do you mean that the metals exposure in this study was possibly related to fish consumption?

I like the revisions to Table 3.

What is the difference between Table 3 (main text) and Table 4 (supplemental)? The text says that there is more detail in the supplemental table, but they seem nearly identical.

I like having the study evaluation results in an Excel file. Please consider putting the two supplemental tables into separate tabs so that the worksheets can be filtered more easily.

In Figure 2, I appreciate you removing the decimals, but now the numbers 0 and 1 are repeated several times. Can you change the legend so that the numbers appear only once?

Author Response

Subject: Resubmission of Manuscript “Neurodevelopmental outcomes associated with early-life exposure to heavy metals: a systematic review”

Dear Dr. Tchounwou,

We hope this message finds you well. We would like to resubmit our manuscript entitled “Neurodevelopmental outcomes associated with early-life exposure to heavy metals: a systematic review” for consideration by the International Journal of Environmental Research and Public Health. We sincerely appreciate the time and dedication of the reviewer in providing constructive feedback.

We have carefully considered all comments and suggestions provided throughout the entire review process. All modifications are clearly highlighted in red in the revised version. Below, we provide a detailed, point-by-point response to each of the reviewer’s comments.

We appreciate the opportunity to resubmit our manuscript and hope it will be considered for publication.

Sincerely,

________________________________________________________________________

 Reviewer 2

I had commented on the original submission that the number of studies identified in the review was rather small, likely due to the lack of specific search terms. The authors responded: “The decision to use more general search terms was intentional and aligned with the research question, which aimed to broadly map the literature to assess whether certain heavy metals would appear more frequently even without explicitly targeting them in the search strategy.” This is fine, but the authors should acknowledge this as a limitation of the review. The identified studies should not be considered a comprehensive overview of the subject.

We appreciate this comment. We have revised the limitations section to explicitly include this information, as follows:

 “The search strategy used may limit this review, as it employed general terms without specific descriptors for each heavy metal. This approach aimed to verify whether certain metals appeared more frequently in the literature. However, it may have restricted the number of studies identified, and thus, the results should not be considered a comprehensive overview of the scientific production on the subject.”

The revised manuscript talks about the results in terms of small, medium, and large effect sizes, which is not something that was in the original draft. The thresholds for small, medium, and large are unclear and need to be defined by the authors. I think it could be useful to talk about effect sizes, but I do not think that current organization of the text makes much sense; e.g., lead and mercury appear in all three sections (small, medium, and large). Consider reorganizing the discussion of effect sizes into a short paragraph or maybe adding a figure that summarizes the effect size findings for different metals.

We appreciate the comment and acknowledge that this description was not present in the original version of the manuscript. We clarify that the current format was developed to address the reviewers’ requests. Effect size thresholds are determined by the statistical methods used to correlate metal levels with each neurodevelopmental outcome in each study. Consequently, the same metal may exhibit different effect sizes (small, medium, or large), depending on the outcomes assessed and the strength of the associations found. This variation can occur both within the same study and across different studies.

While we recognize that a more concise approach could be useful, condensing all information into a single paragraph could obscure the detailed presentation of associations, effect sizes, and study quality, which we organized by metal as previously requested.

We also acknowledge that criteria for effect size thresholds were not initially clarified. We addressed this by adding relevant information to the results section and partially incorporated the suggested improvement by grouping some metals within the same paragraph. The modifications include:

“Effect size thresholds are determined by the statistical methods used to correlate metal levels with each neurodevelopmental outcome, as reported in each study. These statistics are detailed in Table 3. Given the variety of outcomes assessed and the different statistical metrics applied across studies, the same metal may show associations of varying magnitudes, small, medium, or large, reflecting heterogeneity in the strengths of associations observed both within the same study and across different studies.”

As, Cd and Mn

Good to fair quality studies have shown that As exposure was associated with worsening cognition, social skills, motor development, increased risk of ASD, and global neurodevelopmental performance [60, 61, 62, 46]. Small effect size associations for Cd indicated increased risk of ADHD and ASD, as well as worsening motor function, observed in good to fair quality studies [61, 40, 60]. Exposure to Mn was associated with increased risk of ASD and poorer cognitive performance in good quality studies [61, 62].”

In Figure 1, consider changing the text in the first screening level to something like “Records screened by title/abstract” and “Records excluded by a human during title/abstract screening”.

We appreciate this comment and have revised the flowchart accordingly to incorporate these modifications.

At the bottom of Figure 1, what does “Indirect association” refer to? Please specify this somewhere.

We appreciate this comment. The term “Indirect association” refers to studies that did not assess a direct association between heavy metal exposure and neurodevelopmental outcomes. We have revised the figure caption to include this clarification.

The methods section talks about study quality ratings as “high, moderate, low”, whereas the results talks about “good, fair, poor” or sometimes “good, fair, low”. Please use consistent terminology.

We appreciate this insightful observation. Although a terminology change was made in response to the previous round of review, standardization of quality-related terms throughout the manuscript was incomplete. To address this, we have relocated the quality assessment results to subsection 3.3 and carefully reviewed the text to ensure consistent use of the terms ‘good quality,’ ‘fair quality,’ and ‘low quality.’

Line 409: “However, in our review, Kou et al. [77] identified negative effects of combined exposure to Cd, Hg, and Pb on language development, possibly exacerbated by fish consumption.”  Wording is unclear; do you mean that the metals exposure in this study was possibly related to fish consumption?  

We appreciate this comment and acknowledge that the original wording was unclear. As diet was not the focus of the cited study, we have removed this reference to avoid introducing secondary details that could potentially confuse the interpretation of results.

I like the revisions to Table 3.

What is the difference between Table 3 (main text) and Table 4 (supplemental)? The text says that there is more detail in the supplemental table, but they seem nearly identical.

We appreciate this insightful observation. Table 4 includes additional details that, due to space limitations, could not be incorporated into the main text. These differences comprise: (1) the objectives of the selected studies and (2) specific information regarding metal analysis, such as exposure, analytical methods used, and the metals analyzed.

I like having the study evaluation results in an Excel file. Please consider putting the two supplemental tables into separate tabs so that the worksheets can be filtered more easily.

We appreciate this feedback. To facilitate easier review by readers, we will place the two supplemental tables on separate tabs within the Excel file.

In Figure 2, I appreciate you removing the decimals, but now the numbers 0 and 1 are repeated several times. Can you change the legend so that the numbers appear only once?

Thank you for this question. The figure has now been updated so that each number appears only once in the legend.

We sincerely thank the reviewer for the valuable comments and constructive feedback, which have greatly contributed to improving the quality and clarity of our manuscript. We hope that all suggestions have been addressed satisfactorily.
